# Water-mass Evolution in the Cretaceous Western Interior Seaway of North America and Equatorial Atlantic

James S. Eldrett[1], Paul Dodsworth[2], Steven C. Bergman[3], Milly Wright[4], Daniel Minisini[5]

[1]Shell International Exploration & Production B.V, Kesslerpark 1, 2288 GS Rijswijk, Netherlands.

[2]StrataSolve Ltd, 42 Gaskell Street, Stockton Heath, Warrington, WA4 2UN, UK.

[3]Formerly Shell International Exploration and Production Inc, 3333 Highway 6 South, Houston, TX, 77082, USA, currently 20625 Chautauqua Beach Rd. SW, Vashon, WA 90870, USA.

[4]Chemostrat Inc., 3760 Westchase Drive, Houston, Texas, TX 77042. USA.

[5]Formerly Shell International Exploration and Production Inc, 3333 Highway 6 South, Houston, TX, 77082, currently 200 N. Dairy Ashford, Houston, TX, 77079

*Correspondence to* James S. Eldrett (james.eldrett@shell.com)

**Abstract.** The Late Cretaceous Epoch was characterized by major global perturbations in the carbon cycle, the most prominent occurring near the Cenomanian-Turonian (CT) transition marked by Oceanic Anoxic Event/OAE-2 at 94.9 – 93.7 Ma. The Cretaceous Western Interior Seaway (KWIS) was one of several epicontinental seas in which a complex water-mass evolution was recorded in widespread sedimentary successions. This contribution integrates new data on the main components of organic matter, geochemistry, and stable isotopes along a North-South transect from the KWIS to the equatorial western Atlantic and Southern Ocean. In particular, cored sedimentary rocks from the Eagle Ford Group of West Texas (~90-98 Ma) demonstrate subtle temporal and spatial variations in paleoenvironmental conditions and provide an important geographic constraint for interpreting water-mass evolution. High latitude (boreal-austral), equatorial Atlantic tethyan and locally sourced Western Interior Seaway water-masses are distinguished by distinct palynological assemblages and geochemical signatures. The northward migration of an equatorial Atlantic tethyan water-mass into the KWIS occurred during the early-middle Cenomanian (98-95 Ma) followed by a major re-organization during the latest Cenomanian-Turonian (95-94 Ma) as a full connection with a northerly- boreal water-mass was established during peak transgression. This oceanographic change promoted de-stratification of the water column and improved oxygenation throughout the KWIS and as far south as the Demerara Rise off Suriname. In addition, the recorded decline in redox-sensitive trace metals during the onset of OAE-2 likely reflects a genuine oxygenation event related to open water-mass exchange and may have been complicated by variable contribution of organic matter from different sources (e.g. refractory/terrigenous material), requiring further investigation.

Author Contributions: P.D. conducted palynological analyses; M.W. principal investigator for trace metal analyses; D.M. principal investigator for core description; J.E., S.B. and D.M, integrated and analysed the datasets; all authors co-wrote the paper.

The authors declare no conflict of interest.

KEYWORDS: Cenomanian, Turonian, palynology, trace metal, water-mass, boreal, tethyan, OAE, Western Interior Seaway

## 1. Introduction

The Late Cretaceous Epoch was characterized by sustained global warming, emplacement of several Large Igneous Provinces (LIPs), global extinctions, global sea-level highstands leading to several epicontinental seaways, and major global perturbations in the carbon cycle termed Oceanic Anoxic Events (OAE's), the most prominent occurring at the Cenomanian-Turonian transition, and termed OAE-2 (Schlanger and Jenkins, 1976). This event is globally recognized by a positive carbon isotope excursion (CIE) reflecting the widespread sequestration of $^{12}$C-enriched organic matter in marine sediments under global anoxic conditions (see Jenkyns, 2010 and references therein). Proposed hypotheses for initiation of global anoxia and enhanced carbon sequestration include long term triggers such as changes in ocean circulation and eustatic sea level rise, flooding large areas of continental shelves promoting global stratification and stagnation in greenhouse climates (Erbacher et al. 2001), abrupt episodes of volcanogenic activity and emplacement of large igneous provinces (LIPs; Orth et al. 1993; Snow et al. 2005; Turgeon and Creaser, 2008; Kuroda et al. 2007; Duvivier et al. 2014) releasing large quantities of $CO_2$ in the atmosphere and increasing delivery of hydrothermally-derived and weathered nutrients into the photic zone enhancing primary production (*e.g*., Adams et al., 2010); or a combination of both (*e.g.,* Kidder and Worsley, 2010). However, increasing evidence indicates a decoupling in the precise timing of the CIE (hence OAE-2) and the location of organic-rich sediment deposition, reflecting that deposition of organic-rich sediment was modulated and ultimately dependent on local and regional processes (basin restriction, water stratification, bottom currents, sediment input) although favoured by global phenomena (sea level change, orbital forcing) (*e.g*., Trabucho-Alexandre et al., 2010). This is particularly apparent within shallow epicontinental seaways such as the southern Cretaceous Western Interior Seaway (KWIS) where parts of the seaway experienced anoxia and recorded organic-rich black shales prior to the OAE-2 interval, and in contrast to other globally recognized sections that recorded organic-rich sediments during the OAE-2 interval (e.g. Plenus Marl and Bonarelli intervals in Europe); the KWIS recorded relatively organic-lean and oxygenated sediments (*e.g*, Meyers, 2007; Eldrett et al., 2014). Thus, the oceanographic regime of the southern KWIS and its influence on the geologic expression of OAE-2 in this shallow epicontinental sea has been considered unique.

It has been proposed that ocean circulation in the KWIS was restricted during the Cenomanian, promoting anoxia due to a sill in the southern gateway (i.e. Texas/Mexico); and that late Cenomanian sea level rise (Greenhorn cyclothem of Kauffman, 1977, 1984) was sufficient to reach a critical sill depth allowing a breach of the southern end of the seaway allowing rapid incursion of warm, normal saline tethyan waters (Arthur and Sageman, 2005). Previous publications have characterized the inflow of tethyan waters at this time by the i) improved environmental conditions as indicated by the sharp increase in abundance and diversity of foraminiferal/molluscan and ammonite assemblages reaching far north into the KWIS (McNeil and Caldwell, 1981; Kauffman, 1984, 1985; Eicher and Diner, 1985; Elder, 1985; Leckie et al., 1998; Caldwell et al., 1993; Kauffman and Caldwell, 1993, Elderbak and Leckie, 2016) and ii) the lithologic transition from organic-rich mudrocks to a highly bioturbated limestone dominated facies (Corbett et al. 2014; Lowery et al. 2014).

The inflow of southern more saline waters into the southern KWIS during the latest Cenomanian has been proposed to either promote overturning of the water-column or mixing with a northern water-mass resulting in "caballing" and the production of a third more dense water-mass (Hay et al., 1993; Slingerland et al., 1996). In either scenario, the inflow of tethyan water during OAE-2 was interpreted to have cumulated in the abrupt oxygenation of the seafloor as recorded by development and persistent abundance of benthic fauna (i.e. Elderbak and Leckie, 2016). However, this interval of benthic faunal abundance in the KWIS was originally termed the benthonic zone by Eicher and Worstell, (1970) who demonstrated that the benthonic foraminifera zone was best expressed in northerly sections where it spanned the entire Cenomanian-Turonian Bridge Creek Limestone, and is less developed in the central KWIS sections where it is stratigraphically restricted to the uppermost Cenomanian (i.e. beds 68-78 at Rock Canyon, Pueblo, Colorado; Eicher and Worstell, 1970), where it spans only part of the OAE-2 interval and was subsequently termed the Benthic Zone by Keller and Pardo (2004). However, it should be noted that the benthonic/benthic zone in the KWIS has also been shown to correspond with the equatorial migration of boreal dinoflagellate cyst (dinocyst) taxa (Eldrett et al. 2014; Van Helmond et al. 2016) and has been correlated with a short-lived climate cooling episode termed the Plenus Cold Event (PCE; Eldrett et al. 2014; Van Helmond et al. 2014; 2016; Elderbak and Leckie, 2016), whereby similarly cool boreal waters invaded northern and central Europe (Jefferies, 1963; Gale and Christensen, 1996; Voigt et al., 2006; Jarvis et al., 2011) and equatorial waters cooled by up to 4°C (Forster et al. 2007). It is therefore difficult to reconcile the late Cenomanian northerly inflow of a warm tethyan water-mass into the southern KWIS, at a time of global cooling, the southerly restriction in benthic fauna and coeval equatorial migration of boreal taxa and associated water-mass in the KWIS and Europe. It is plausible that a much more complex oceanographic system existed in the KWIS; such as that modelled by Slingerland et al. (1996) and Kump and Slingerland (1999), whereby a strong cyclonic gyre developed in the central KWIS drawing both tethyan waters northward along the eastern margin and boreal waters southward along the western margin of the seaway (see also discussions in Elderbak and Leckie, 2016).

In order to better understand and constrain the nature and timing of water-mass evolution in the southern gateway to the KWIS, and the associated paleoenvironmental and paleoclimatic processes, this contribution

presents detailed palynological and geochemical analyses within a multidisciplinary framework from the Eagle Ford Group (Gr.) and bounding formations of the Buda Limestone and Austin Chalk from southwest Texas, USA. Furthermore, to place the southern gateway of the KWIS into a more supra-regional understanding we also analysed correlative materials from the central KWIS (Portland-1 core, Colorado) and to the south in the Tropical Atlantic (ODP Leg 207, Demerara Rise) and Southern Ocean (ODP Leg 183, Kerguelen Plateau).

## 1.1.    Geological Setting

The Eagle Ford Gr. was deposited during the Cenomanian to Turonian across the broad Comanche Platform in the southern intersection of the KWIS and northern Gulf of Mexico (**Figure 1**), which represented part of the >3,000 km long foreland basin that formed behind the greater Cordilleran retro-arc fold-and-thrust belt during Late Mesozoic through Eocene times along the inboard side of the Cordilleran magmatic arc and accreted allochthonous terranes of North America (e.g., Burchfiel et al. 1992; Dickinson, 2004). The regional tectonic setting was influenced during the Cretaceous by the subduction of the conjugate oceanic plateau to the Shatsky Rise (Liu et al., 2010) dynamic topography from the subducting Farallon Plate (Liu, 2014) and thermally subsiding Gulf of Mexico margin, resulting in the development of broad ramp-shelves including the Comanche Platform with reactivated basement structures defining intra-shelf basins, such as the Maverick Basin and structural highs such as the Terrell Arch. The Cenomanian-Turonian Trans-Pecos region (**Figure 2**) was deposited in a distal sediment starved setting >500 km from the nearest shoreline during a locally quiescent tectonic period resulting in stable platform conditions and gradual subsidence ideal for the preservation of mudstones, limestones and bentonites of the Eagle Ford Gr., the underlying Buda Limestone and overlying Austin Chalk.

## 1.2.    Previous Palynological Studies

Previous Cenomanian-Turonian palynological studies of the KWIS include those of Brown and Pierce (1962), Christopher (1982), Courtinat (1993), Li and Habib (1996), Cornell (1997), Dodsworth (2000; 2016), Harris and Tocher (2003), Eldrett et al (2014, 2015a, b) and Van Helmond et al. (2016). Brown and Pierce (1962) first reported dinocysts and other palynomorphs from the Eagle Ford Gr. in northeast Texas, whereas terrestrial sporomorphs were recovered from the Woodbine interval of the Eagle Ford Gr. by Christopher (1982). Subsequently, most studies have focused on the central part of the KWIS and in particular the Global boundary Stratotype Section and Point (GSSP, "golden spike") for the base of the Turonian at Rock Canyon, Pueblo, Colorado (e.g., Courtinat, 1993; Li and Habib, 1996; Dodsworth, 2000; Harris and Tocher, 2003). For the southern sector of the KWIS relatively few sections have been published, notably Cornell (1997) reporting diversified mainly gonyaulacacean dinocyst assemblages from the massive limestones of the Buda Formation (Fm.) in Dona Ana County, New Mexico; and more recently Eldrett et al (2014) reporting the incursion of boreal dinocysts in the Eagle Ford Gr. during the OAE-2 CIE interval corresponding with improved bottom-water oxygenation. This equatorial migration of the boreal dinocyst complex *Cyclonephelium compactum–C. membraniphorum* has subsequently been recorded across the KWIS and northern Tethys (see Van Helmond et

al. 2016). Many of the regional and globally recognized age diagnostic biostratigraphic events encountered in the KWIS were calibrated against an astronomically tuned and geochronologically constrained age model for the Cenomanian-Turonian and early Coniacian based on a relatively expanded section of the Eagle Ford Gr. and bounding units of the Austin Chalk and Buda Limestone that were recovered from the Shell Iona-1 research

core, in west Texas (Eldrett et al. 2015a).  Paleoenvironmental reconstructions of the Eagle Ford Gr based on the biostratigraphic assemblage data presented by Eldrett et al. (2015a) were beyond the scope of that paper; but some aspects were presented as part of an integrated multidisciplinary contribution demonstrating that obliquity and precession forcing on the latitudinal distribution of solar insolation may have been responsible for the observed lithological and environmental variations through the Cenomanian, Turonian and Coniacian in mid-

latitude epicontinental sea settings. Subsequently, Dodsworth (2016) provided more detailed paleoenvironmental interpretations from palynological assemblages primarily from the Eagle Ford Gr. exposed at the Lozier Canyon cliff section, Terrell County, west Texas, which is similar to the Innes-1 core (see below) and contains a significant unconformity (duration > 2 million years; Myrs) within the Turonian. This unconformity is much less intense (duration <0.2 Myrs) in the more distal Iona-1 core.

This paper aims to build on and expand these previous contributions by:  i) providing detailed palynological interpretations of the Eagle Ford Gr. based on several locations enabling a more widespread geographic understanding; ii) integrating  multidisciplinary datasets including organic and inorganic geochemistry allowing a greater understanding of the main paleoenvironmental controls; and iii) presenting new data analysed from sections further to the north in the KWIS (Portland-1 core) and to the south in the Tropical Atlantic (ODP Leg

207, Demerara Rise; **Figures 1-2**) and Southern Ocean (ODP Leg 183, Kerguelen Plateau) to provide insights into supra-regional ocean circulation and water-mass evolution during the Cenomanian-Turonian Greenhouse climate state.

### 2.    Material & Methods

Core material from the Eagle Ford Gr., and bounding units in the Maverick Basin, West Texas, U.S.A., were analysed for visual kerogen analyses (palynology, palynofacies).  The core material was sampled along a physiographic transect from the main Comanche carbonate shelf (Innes-1) towards the edge of the Maverick intra-shelf basin (Iona-1) and central part of the intra-shelf basin (well 'X'; **Figure 1-2**). Outcrop sections were also analysed in the San Marcos Arch area near Austin along the Bouldin Creek section. In order to compare

data collected from the Eagle Ford Gr., with any regional trends, data were also collected further to the North near the base of the Turonian GSSP (Kennedy et al. 2005; Sageman et al. 2006) in Colorado (USGS Portland-1) and to the South in the Tropical Atlantic off Suriname (ODP Leg 207; Sites 1260 and 1261, Demerara Rise; Shipboard Scientific Party, 2004a, 2004b). Material was also collected from Site 1138, Kerguelen Plateau (Shipboard Scientific Party, 2000). Palynological analyses on these sites were conducted under both transmitted

and fluorescence microscopy and the results supplemented and compared to newly collected and previously published organic and inorganic geochemical analyses including total organic carbon (TOC) and major, minor and trace elements (Erbacher et al. 2005; Forster et al. 2007; Friedrich et al. 2008; Hetzel et al. 2009; Joo and Sageman, 2014; Duvivier et al. 2014; Lowery et al. 2014; Eldrett et al. 2014; 2015a, b; Sun et al. 2016; Dickson

et al. 2016; Minisini et al. in review). Palynological parameters presented include i) the ratio between terrestrial (T) and marine (M) palynomorphs (T:M ratio) as a proxy for terrestrial input; ii) the ratio between peridinioid or P-cysts and gonyaulacoid or G-cysts (P:G ratio) of the dinocyst assemblage as a proxy of nutrient input. Diversity of the dinocyst assemblage was also calculated using both Shannon-Wiener (H) and Simpson-Hunter (D) indexes. Detailed palynological methods and associated discussion of paleoenvironmental parameters are provided in the supplemental information. In addition to the quantification of organic foraminifera test linings from palynological residues, benthic foraminifera abundances were also counted from a combination of micropaleontological picked residues and thin sections following the methods detailed in Eldrett et al. (2015a) and supplemented by published benthic foraminiferal records (e.g. Friedrich et al. 2006, 2008, 2011). Principal Component Analyses (PCA) were run on this integrated palynological-geochemical dataset to elucidate the primary controls for paleoenvironmental and paleoclimatic interpretations. Details of site locations and methodologies employed are presented in the supplemental information.

## 3. Results

The following section describes some of the key results that are referred to in the discussion section and presented in **Figures 3-11**. Detailed results and data are presented in the supplemental information and datafile.

### 3.1. Organic carbon-isotope Stratigraphy

Significant organic carbon isotopic differences are observed throughout the studied interval. The Buda Limestone record $\delta^{13}C_{org}$ values ~-26‰ and become more negative within the Eagle Ford Gr. where background values are ~-27 to -28‰ for Iona-1, Innes-1 and well 'X'. From these background levels, five notable positive-; and three negative- carbon isotope excursions (CIE) of varying magnitudes were recognized. These excursions can be correlated with the English Chalk reference section of Jarvis et al. (2006; see Eldrett et al. 2015a for detailed discussion of biostratigraphic calibration of CIE's in the KWIS) and compared with the previously published data for ODP Leg 207 as presented in **Figure 3**. Two of these CIE's have specific relevance for this contribution: i) the ~2‰ positive $\delta^{13}C_{org}$ excursion in Iona-1 (143.73m-139.27m) and Innes (76.88m-74.63m) that corresponds with the Middle Cenomanian Event (MCE) and ii) the Cenomanian-Turonian CIE that is clearly expressed with a positive CIE of up to 4‰ occurring in Iona-1 (112.45m-92.73m), Innes-1 (55.74 m- 42.51 m) and well 'X' (1639.9 m – top not recorded as above the cored interval). It should be noted that the definition of the base of the Cenomanian-Turonian CIE has been re-interpreted to include the precursor events presented in Eldrett et al. (2014, 2015a); as such the base of the CIE at Iona-1 is moved from 105.96m to 112.45m and is assigned an age of 95.01±0.12 Ma based on the obliquity age model presented in Eldrett et al. (2015).

### 3.2. Geochemistry

#### 3.2.1. Southern KWIS (Texas)

The Buda Limestone (early Cenomanian; ca 98-97.5 Ma) is characterized in the Iona-1 core by high percentage of Calcium Oxide (CaO, >45 wt %) and low percentages of Aluminium Oxide ($Al_2O_3$: <2 wt %), Silicon Dioxide ($SiO_2$: <6 wt %) and Titanium Dioxide ($TiO_2$: <1 wt %). Exceptions occur in the interbedded bentonite layers that record higher values of $Al_2O_3$ (>10 wt %); $SiO_2$ (>25 wt %) and $TiO_2$ (~1 wt %). Redox sensitive trace metal concentrations and enrichment factors (EF) in the limestones are low with Molybdenum (Mo); Uranium (U) and Vanadium (V) recording <1ppm/<2 EF; <2ppm/<4 EF and <20ppm/<2 EF respectively. The lower Eagle Ford Gr. (ca. 97.2-94.9 Ma) is characterized by high TOC and concentrations of redox sensitive trace metals that are enriched compared to average shale. In particular, the basal most part of the lower Eagle Ford Gr. at Iona-1 (153m – 144m) records the highest trace metal enrichments with $Mo_{EF}$ = ave. 400, $U_{EF}$ = 12 and $V_{EF}$ = 17. These trace metal enrichments decline sharply at the base of the MCE interval (143.37m – 142.27m) after which they increase again and become relatively enriched ($Mo_{EF}$ = ave. 140, $U_{EF}$ = 7 and $V_{EF}$ = 13). A similar trend is also recorded in both Innes-1 and well 'X' albeit at overall lower values. In all three cores redox sensitive trace metal concentrations and enrichments start to decline prior to the lower - upper Eagle Ford Gr. boundary (~94.6 Ma). The upper part of the Eagle Ford Gr. is characterized by generally low redox sensitive trace metal concentrations and enrichments in all three cores, with minima associated with the interpreted benthic oxic zone (see Eldrett et al. 2014). However, within the Benthic Oxic Zone, there is a recorded increase in Ti/Al, maffic trace elements and europium anomaly (Eu/Eu*; see Eldrett et al. 2014; Figure DR5). Variations in redox trace metals occur between the limestone-marlstone couplets; with marlstones recording slightly elevated enrichments compared to the limestones (see Eldrett et al. 2015b for details). The uppermost part of the Eagle Ford Gr. (corresponding with the Langtry Member; *sensu* Pessagno, 1969; Mb. ~92.5- 90.3 Ma) record a slight increase in redox sensitive trace metals in mostly marlstone lithologies in both Innes-1 and Iona-1 cores (**Figures 4-5**). The overlying Austin Chalk is generally not enriched in redox sensitive trace metals with the exception of the interbedded marlstones. The Bouldin Creek outcrop locality was not analysed for trace metals as part of this study.

#### 3.2.2. Central KWIS (USGS Portland-1, Colorado)

The Dakota Sandstone and Graneros Shale interval in the USGS Portland-1 core (213m to 175m; ~100-95.8 Ma; **Figure 8**) is predominantly comprised of high relative concentrations of $SiO_2$ (ave. ~60 wt %) and $Al_2O_3$ (ave. ~15 wt %). The Thatcher Limestone interval (~185m; ~96.5 Ma) is represented by a single sample that records a slight relative increase in CaO (~8 wt %), but it is likely that this sample adequately represents or resolves the varied lithological end-members recorded in this interval. A relative increase in CaO is recorded from the base of the Lincoln Shale Mb. to near the top of the Hartland Shale Mb. (175.29m – 150m; 95.8- 94.7Ma) reaching up to 25 wt %; and then increase substantially to ~40 wt % CaO marking the lithological break of the base of the Bridge Creek Limestone Mb. (~148.6m).

Throughout the Dakota Sandstone and Graneros Shale intervals the majority of the redox sensitive trace metals record low concentrations and are not significantly enriched compared to average shales (e.g. U, V, Ni, Cu, Zn) and Mo is only slightly enriched ($Mo_{EF}$ = ave. 3; ppm= ave. 4 ppm; **Figure 8**). Redox sensitive trace metal concentrations increase slightly in the Lincoln Shale Mb. and only become significantly enriched in the Porltand-1 core during the deposition of the Hartland Shale Mb. The overlying Bridge Creek Limestone Mb. and in particular the interval recording the OAE-2 CIE is characterized by an overall reduction in redox sensitive trace metals and lack of significant enrichments compared to average shale values. However, variations in redox sensitive trace element enrichments are recorded in the marlstone and limestone couplets of the Bridge Creek Limestone Mb.

### 3.2.3. Demerara Rise, Atlantic Ocean (ODP Leg 207, Sites 1260 and 1261).

The pre-OAE-2 sediments recovered from ODP sites 1260 and 1261 record high and variable values of CaO (25-50 wt %) which decline substantially during the OAE-2 interval as $SiO_2$ values increase with maximum values of 47 wt % and 23 wt % recorded for each site respectively (see supplemental datafile). At Site 1260 in the lower part of the studied interval (445.19 and 462.7 meters composite depth; m.c.d) high concentrations of redox sensitive trace metals are recorded that are significantly enriched compared to average shale (see supplemental datafile; **Figure 9**). Concentrations and enrichments of the redox sensitive trace metals decline from these high values around ~443 m.c.d and reach minima during part of the OAE-2 CIE interval (426-423 m.c.d), but still remain enriched compared to average shale ($U_{EF}$ = 10; $Mo_{EF}$ = 25; $V_{EF}$ = 4). This same trend is mirrored in the sedimentary record from Site 1261; with minima enrichment values recorded during the OAE-2 CIE interval (albeit in lower resolution) that even though are relatively low compared with the pre-OAE-2 interval are still enriched compared to average shale (minima values: $U_{EF}$ = 12; $Mo_{EF}$ = 46; $V_{EF}$ = 5; **Figure 10**).

### 3.3. Palynology

Detailed palynological results and data tables are presented in the supplemental dataset and displayed in **Figures 4-11**.

## 4. Discussion

### 4.1. Principal Component Analyses (PCA)

In order to provide additional understanding into the main controls on organic matter composition and paleoenvironmental significance, PCA was initially conducted on the palynological and geochemical datasets separately and then on the integrated palynological and geochemical dataset. All three analyses showed the similar principal components and clusters and in support of the discussion only the PCA results run on the combined dataset are presented. The combined dataset PCA was run on cores that had multiple detailed analyses from the same core sample/depth, including Innes-1, Iona-1, well 'X', USGS Portland-1 and ODP Sites 1260 and 1261 using the statistical software C2 (http://www.staff.ncl.ac.uk/stephen.juggins/software/C2Home.htm). Variables within the dataset include palynology, major and trace elements, $\delta^{13}C_{org}$ and TOC. The PCA results show a clear grouping of samples along the two primary axes (**Figure 12**).

Eight groups were identified and are lithostratigraphically defined: Group I = lower Eagle Ford; Group II = upper Eagle Ford; Group III = Buda Limestone and Austin Chalk; Group IV = Bridge Creek Limestone; Group V= Hartland Shale Mb.; Group VI =Graneros Shale; Group VII = Dakota Sandstone and Group VIII = Demerara Rise. The groups are discussed in a chronostratigraphic context in Section 4.2. Along the principle axis/eigen score 1, samples with high negative scores comprise high CaO contents and are associated with the Buda Limestone and Austin Chalk samples, whereas high positive scores correspond with high values of elements enriched in heavy minerals (Zirconium-Hafnium; Zr-Hf), silicates (quartz, feldspar), phyllosilicates/clay minerals (Gallium-Potassium Oxide-Rubidium-Titanium; Ga-$K_2O$-Rb-Ti [eg., illite, biotite, smectite, kaolinite], and increased terrigenous contributions (T:M ratio) locally corresponding with the clastics of the Dakota Sandstone Gr. The high positive score along axis 1 may not solely represent terrestrial/detrital riverine dilution, but may also reflect diagenetic alteration of the abundant volcanic ash from atmospheric fallout of western Cordillera plinian eruptions ($SiO_2$, $Al_2O_3$, $TiO_2$, heavy minerals) that were transformed into smectite–illite-Fe-Ti oxides (see Eldrett et al. 2015b). Therefore, the clustering of the environmental variables along the Axis/Eigen score 1 is interpreted as representing the carbonate - non carbonate/volcaniclastic mixing trend.

Axis/Eigen score 2 is interpreted to represent a restricted/eutrophic/anoxic to open marine/oligotrophic/oxic marine trend influencing the water column and/or sediment-water interface; with high positive scores associated with high values in TOC, redox sensitive trace metal concentrations, preservation of amorphous organic matter (AOM) and assemblages dominated by prasinophyte phycomata, the latter are indicative of eutrophic and stratified water column conditions (cf. Prauss, 2007)

It is also noted that high eigen scores are associated with dinocyst assemblages dominated by peridinioid (P-) cysts and particularly taxa comparable to those informally described from Tarfaya, Morocco by Prauss (2012a, b); *Bosedinia cf.* sp 1 & 3 of Prauss (2012b) (photographic illustrations are provided in the supplemental information). Sporadically common/abundant occurrences of *Bosedinia* sp. 1 & sp. 3 were recorded in the Cenomanian and Turonian of Tarfaya by Prauss (2012a, b), where it is often associated with common occurrences of the colonial green alga *Botryococcus*; high concentrations of *Botryococcus* had previously only been reported from fresh/brackish-water paleoenvironments (e.g. Zippi, 1988) and this led Prauss to infer a freshwater affinity for the *Bosedinia*, suggesting it probably represents episodic salinity stratification in the

marine paleoenvironment at Tarfaya (Prauss, 2012c).  However, in all the localities investigated in this study and also Lozier Canyon (Dodsworth, 2016), abundances of *Bosedinia* cf. sp 1 & 3 are not associated with an increase in *Botryococcus* or terrigenous pollen and spores, or with the presence of fresh-water algal genera such as *Concentricystes, Ovoidites* (*Schizophacus*)*, Pediastrum* and *Schizosporis*. The latter genera are common components at Cenomanian – Turonian continental to fluvial-deltaic settings in Utah (Akyuz et al., 2016) and east Texas (Dodsworth, 2016). In the present study, the persistent occurrence of these freshwater algal genera are restricted to the Dakota Sandstone in the Portland-1 core where they are associated with the only frequent occurrences of *Botryococcus* and super-abundant (>50%) occurrences of terrigenous spores and pollen reported here. Influxes of *Bosedinia* do not co-occur in any of these proximal KWIS settings. The PCA results presented in this study indicates that *Bosedinia cf.* sp 1 & 3 plots positively along eigen score 2, associated with indicators for eutrophic and anoxic marine conditions; potentially representing the enhanced nitrite/nitrate availability in the photic zone (see Dodsworth, 2016).  During the Cenomanian Epoch, abundances of *Bosedinia cf.* sp 1 & 3 are confined to middle-lower latitudes (Prauss 2012a, b; this study), being absent from higher latitude records (e.g. Foucher, 1980; Jarvis et al. 1988, 2011; Fitzpatrick, 1995; Pearce et al. 2003, 2009; Lignum 2009), and is thus interpreted to have a tethyan equatorial Central Atlantic affinity (see supplemental information).

The high negative eigen scores along axis 2 correspond with proxies associated with open marine oligotrophic conditions such as gonyaulacoid dinocysts (G-cysts) including *Spiniferites* spp., and *Pterodinium* spp.; the latter shows affinity with modern-day *Impagadinium* spp. which is found in oceanic and open marine conditions (Wall et al. 1977; Zonneveld et al. 2013). High negative eigen scores are also associated with proxies indicative of more oxygenated water conditions such as higher concentration of the redox sensitive trace metal Manganese Oxide (MnO) and foraminiferal test linings. The negative eigen scores are therefore interpreted as representing an open marine, oxygenated and oligotrophic depositional environment.   In addition, it is interesting to note that although phytoclasts plot negatively along eigen axis 2 and may represent a reduced masking effect of AOM during oxygenated conditions (Tyson, 1995); they also plot positively along Eigen score 1 (non-carbonate/volcaniclastic trend) alongside freshwater algae and Areoligeracean dinocysts; the latter are more suggestive of a more nearshore environment (see Brinkhuis and Zachariasse 1988; Harker et al. 1990; Li and Habib 1996).  In addition, a component of this trend may also represent recycling and transportation of nearshore palynomorphs to relatively distal environments. The eigen scores are plotted against depth for each of the sites (**Figures 4-6, 8-10**) to further support the paleoenvironmental interpretation as discussed below.

## 4.2.    Paleoenvironmental Interpretation

### 4.2.1.    Southern KWIS

The Buda Limestone (early Cenomanian; ca 98-97.5 Ma) comprises highly diverse and low abundance dinocyst assemblages (see also Cornell, 1997) indicative of open marine oligotrophic conditions. This interpretation is supported by the i) PCA results reflecting the lack of enrichment in redox sensitive trace metals, low TOC and poor preservation of AOM; dominance of G-cysts and ii) sedimentological and micropaleontological evidence

for abundant and diverse benthos indicative of a healthy carbonate factory. The Buda Limestone dinocyst assemblages are comparable to those reported from marine limestone facies of early Cenomanian age in western Europe, including England (e.g. Cookson and Hughes, 1964) and France (e.g. Foucher, 1980).

The overlying lower Eagle Ford Gr. interval (ca. 97.2-94.9 Ma) is generally characterized by a decline in dinocyst species diversity, with the palynological assemblages comprising high absolute abundance of prasinophyte phycomata with P-cysts being the major component of the dinocyst community indicative of eutrophic and stratified water column conditions (cf. Prauss, 2007; Sluijs et al. 2005). This interpretation is supported by PCA results reflecting the enrichment and co-variance in redox sensitive trace metals and high

TOC values combined with low, but sporadic occurrences of benthic foraminifera indicative of restricted and suboxic-anoxic depositional conditions (see also Eldrett et al. 2014). Furthermore, the presence of aryl isoprenoids in the lower Eagle Ford Gr. section from Iona-1 has been demonstrated as originating from *Chlorobi* (green sulphur bacteria) and thus evidence for at least temporary and/or partial photic zone euxinia at this time (Sun et al. 2016).  Bed- scale variations are also identified in redox conditions recording greater water-mass

ventilation and current activity during the deposition of limestone beds compared to deposition of marlstone beds due to combined obliquity and precessional forcing on solar insolation (Eldrett et al. 2015b). Comparable palynological assemblages have not been reported from the middle to upper Cenomanian deposits in Europe but high relative abundances of prasinophyte phycomata and the peridinioid genus *Bosedinia* have been documented further south in the Cenomanian and Turonian organic-rich shale facies at Tarfaya in north Africa (Prauss 2012a,

2012b).

Within the middle part of the lower Eagle Ford Gr. (~96.1-95.4 Ma) the relative abundance contribution of prasinophyte phycomata to the palynological assemblage decreases as the absolute abundance of the P-cyst *Bosedinia cf.* sp 1 & 3 significantly increases. This shift in the palynological assemblage is recorded in all the studied sections including the San Marcos Arch outcrop; also in the nearby Lozier Canyon outcrop section

(Dodsworth, 2016).  As discussed in **section 4.1**, the abundance of *Bosedinia* sp. 1 & sp. 3 had previously been interpreted as reflecting the occurrence of freshwater/brackish water conditions in the photic zone at Tarfaya and episodic salinity stratification there (Prauss 2012a, 2012b, 2012c), but in the absence of freshwater algae and the combined PCA results for the studied sections this interpretation is not supported here for the KWIS or Demerara Rise. Alternatively, Dodsworth (2016) proposed that the introduction of waters from respectively

deeper denitrification zones into photic zone by vertical expansion of the oxygen-minimum zone or by upwelling may be a controlling factor, with reduced nitrogen/ ammonium favouring prasinophyte algal production (cf. Prauss 2007) and availability of nitrite/nitrate promoting P-cyst (e.g. *Bosedinia* sp. 1 & sp. 3) productivity in the surface waters.

The upper Eagle Ford Gr. associated with first peak ("A") of the OAE-2 CIE (~94.65 Ma) is characterized by a

sharp change in palynological assemblages with increase abundance of G-cysts, in particular open marine forms including *Pterodinium*, *Spiniferites ramosus* and *Nematopshaeropsis* spp. Prasinophyte phycomata become rare and overall dinocyst diversity increases. This assemblage shifts along with the reduction in redox sensitive trace metal enrichments, lowered TOC, increased bioturbation and occurrence of benthic foraminifera indicate

deposition within an open marine meso-oligotrophic and oxygenated depositional environment (also see Eldrett et al. 2014). By contrast, in some European depositional basins, e.g. eastern England, northern Germany and Crimea, uppermost Cenomanian to lower Turonian deposits associated with OAE-2 are characterized by intervals of interbedded organic-rich mudrocks with limestones, the former contain isolated influxes of prasinophyte phycomata and higher numbers of P-cysts than under- and overlying formations (Marshall and Batten, 1988; Dodsworth, 1996; 2004; Prauss, 2006); whereas limestone deposition, lean in organic matter, is continuous in other areas, e.g. southern England – northern France. The relative increase in the T:M ratio during OAE-2 in the studied sections presented here likely reflects the closed-sum effect as absolute abundance of marine palynomorphs decreases during the upper Eagle Ford (transition from eutrophic to meso-oligotrophic conditions). Absolute abundance of terrestrial palynomorphs does not increase significantly, although this may partly reflect greater distance from shoreline during the Cenomanian-Turonian transgression and potentially diluted concentration of all palynomorphs including pollen during increased biogenic carbonate productivity associated with the limestones of the upper Eagle Ford – Lower Bridge Creek. An increase in absolute abundance of terrestrial palynomorphs is recorded at the Pueblo GSSP, in uppermost Cenomanian beds 82-lower 85 (2000-5000 counts per gram; 30-60%; Dodsworth, 2000). It is unclear whether palynological assemblages in the studied sections support increased hydrological cycle during OAE-2 (see Van Helmond et al. 2014). The sporomorph assemblages during OAE-2 mainly record a relative increase in gymnosperms, in particular during the PCE interval, and thus any increase in T:M ratio may reflect transition from mega-thermal to meso-thermal vegetation (perhaps also reflecting increased pollen production by wind dispersed gymnosperms) in response to climate cooling episode (see Forster et al. 2007; Jarvis et al. 2011) rather than increased hydrologic cycle. The recorded increase in Ti/Al in the PCE interval may reflect increased fluvial and/or eolian inputs, however the association of Ti/Al with trace metal enrichments in Cobalt, Chromium, Scandium as well an increase in heavy-to-light rare earth elements and a positive europium anomaly, together are indicative of a hydrothermal or mafic influence and suggest emplacement and weathering of a LIP during the PCE interval (see Eldrett et al, 2014). In addition, the Ti/Al record may also reflect alteration of biotite from the ubiquitous felsic volcanic ash beds rather than solely eolian versus fluvial inputs, and the links with gymnosperm abundances could have resulted from changes in climate and/or oceanographic conditions resulting from these large scale and regional igneous events.

The PCA results indicate occasional dysoxia-anoxia within the upper Eagle Ford Gr. interval; which becomes more persistent in the upper-part of the Langtry Mb. as indicated by slight enrichment in redox sensitive trace metals and palynological assemblages recording reduced dinocyst diversity and enhanced abundances of prasinophyte phycomata. This interpretation seems to contrast with the sedimentological evidence that preserves bioturbated marlstones and nodular limestones along with shell hash horizons forming symmetric ripples and abundant in situ macrofauna (i.e. echinioids); all indicative of a high energy dynamic environment (Minisini et al. in review). It may be the case that short-lived storm and oxic events are not resolved at our sampling resolution as bulk geochemical and palynological data intergrates longer time periods; or alternatively we resolve short-lived periods of dysoxia-anoxia that are subsequently smeared and re-distributed by bioturbation within a background environment characterized by high energy and well-oxygenated conditions. It should be noted that similar abundance increases in prasinophyte phycomata are recorded in the upper part of the South Bosque Formation at the Bouldin Creek outcrop and in the Lozier Canyon outcrop section (Dodsworth, 2016). The occurrences of prasinophyte phycomata in the Lozier Canyon outcrop were thought to represent an artifact

of their preferential preservation in weathered material (Dodsworth, 2016), however, compared to the regional trends identified in well-preserved core alongside trace metal enrichments suggests a genuine transition to at least partially and/or episodically dysoxic-anoxic depositional environment. The limestones of the Austin Chalk record a return to open marine and persistently oxygenated conditions supported by the PCA results reflecting very diverse dinocyst assemblages, comprised mainly G-cysts with abundant foraminiferal test linings, reduced AOM and no enrichments in redox sensitive trace metals.

### 4.2.2. Regional water-mass evolution

In this study we infer three main water-masses: i) equatorial-Atlantic tethyan source; ii) a northern boreal source and iii) a more local central KWIS source. The equatorial-Atlantic sourced water is primarily based on the occurrence of dinocysts and other microfauna (i.e. calcareous nannofossils and foraminifera) that are geographically restricted to low latitudes and specifically equatorial Atlantic with inferred tethyan affinities (e.g. *Bosedinia* cf. sp 1 & 3; see discussion below). The surface waters of this watermass is characterized by low diversity dinocyst assemblages dominated by P-cysts and prasinophyte phycomata indicative of stratified and hydrographically restricted and eutrophic conditions (this study); whilst the correspondence with the presence of isorenieratene derivatives (i.e. Sinninghe Damsté and Köster, 1998; Kuypers et al. 2002; Kolonic et al. 2005; Van Bentum et al, 2009; Sun et al. 2016) demonstrate at least temporary and/or partial photic zone euxinia. The underlying bottom-water; or at least sediment-water interface was predominantly dysoxic-anoxic as evidenced by the enrichment in redox sensitive trace metals and relatively sparse benthic foraminifera (i.e. Hetzel et a. 2009; Eldrett et al., 2014). The northerly sourced watermass is constrained by the occurrence of dinocysts with a boreal affinity (see Eldrett et al. 2014; Van Helmond et al. 2014; 2016); combined with the increased planktonic diversity and dominance of G-cysts that are associated with more oligotrophic/hydrographically unrestricted conditions. The absence of prasinophyte algae point towards a mixed water-column (cf. Prauss, 2007); and the overall low values in redox sensitive elements and increased abundance and diversity of benthic fauna along with enhanced bioturbation indices (e.g. Meyers et al. 2007; Eldrett et al. 2014) indicate frequent ventilation of the bottom-waters and/or sediment-water interface. A more local and partially restricted dysoxic watermass sourced by the central KWIS is interpreted based on the occurrence of mixed moderate diversity dinocyst assemblages, including taxa more typical of the northern and central KWIS, with rare tethyan components combined with limited enrichment in redox sensitive elements and abundance of low diversity agglutinated benthic foraminfera assemblages (see discussion below). It should be noted that within these regional watermass regimes; higher frequency variations in redox state are documented; in part a response to obliquity and precession forcing on the latitudinal distribution of solar insolation (see Eldrett et al. 2015b). Comparing the paleoenvironmental trends recorded in the Cenomanian – Coniacian sections from SW Texas to those recorded further north within the central KWIS (USGS Portland-1; Pueblo GSSP outcrop) and to the south in the equatorial North Atlantic (ODP sites 1260, 1261, Demerara Rise) allows the reconstruction of water-mass evolution (**Figure 13-14**). A discussion of dinocyst and pollen paleo-latitudinal provinicialism across these regions during Cenomanian and Turonian times is given in the supplemental information.

Lower Cenomanian sediments from the equatorial Atlantic (ODP Site 1260) are interpreted as being deposited in a stratified suboxic-anoxic marine environment as indicated by enrichment in redox sensitive trace metals; low diversity dinocyst assemblages with abundance of prasinophyte phycomata; preservation of laminated organic rich mudrocks and positive PCA-2 scores. This interpretation is consistent with a circulation controlled nutrient trap fuelling surface water productivity and anoxic depositional environment as proposed by Jiménez Berrocoso et al. (2010) and Trabucho-Alexandre et al. (2010). Further to the North in the southern part of the KWIS, the early Cenomanian is characterized by the deposition Buda Limestone, which is primarily characterized by well oxygenated conditions. The first incursion of an equatorial-Atlantic tethyan water-mass into southern Texas occurred at 97.2 Ma (t1; **Figure 14**), marked by the hummocky cross stratified limestones and mass transport deposits interbedded with organic rich sediments that characterize the lowermost Eagle Ford Gr., above the top Buda limestone submarine unconformity (see Minisini et al. in review). In the Central KWIS (Portland-1 core), the early Cenomanian is characterized by relatively dysoxic depositional conditions, with frequent to common prasinophyte phycomata and mixed dinocyst assemblages, including taxa more typical of the northern and central KWIS such as *Senoniasphaera microreticulata* and *Palaeoperidinium cretaceum*, and the first consistent but mainly rare occurrence of *Bosedinia* cf. sp 1 & 3 which is common in coeval deposits at Demerara Rise; along with low diversity agglutinated benthic foraminfera (**Figure 8**) and the occasional occurrence of rare tethyan calcareous planktonic taxa, together suggestive of a mainly Western Interior Seaway source (Eicher and Diner, 1985). The initial northward expression of an equatorial-Atlantic tethyan water-mass is only evidenced in the central KWIS (Portland-1 core) during the deposition of the Thatcher Limestone Mb. (ca 96.5-96.6 Ma), with slight increases in prasinophte phycomata, decrease in dinocyst diversity, reduced benthic foraminifera and supported by the occurrence of ammonites, calcareous nannofossils and an increase in planktonic foraminifera with tethyan affinities (see Eicher and Diner, 1985; Cobban, 1993; Hancock et al., 1993, Bralower and Bergen, 1998). Water-mass characteristics fluctuate during the middle Cenomanian in the Portland-1 core, but are interpreted as being predominately sourced locally from the Western Interior Seaway as evidenced by mixed dinocyst assemblages; relatively low enrichments in redox sensitive trace metals and dominant occurrence of agglutinated foraminifera. This study is limited by the relatively low sampling resolution from the MCE interval; in order to better resolve the detail and timing of these paleoceanographic variations higher resolution multi-proxy analyses are required. In the Portland-1 core there is a clear shift from agglutinated to calcareous benthic foraminfera near the top of the MCE interval; which combined with the increased abundances of calcareous planktonic foraminifera (e.g. *Rotalipora cushmani*) and dinocysts (e.g. *Bosedinia cf.* sp. 1 & 3) with tethyan affinities is suggestive of an equatorial-Atlantic tethyan influence (**Figure 8;** for detailed discussion of tethyan-boreal foraminiferal distribution within Colorado, see Eicher and Diner, 1989). However, it is only during the deposition of the middle part of the Lincoln Shale (~95.6 Ma) that abundant prasinophyte phycomata, increased trace metal enrichments and reduced dinocyst diversity point towards a persistent influence of an equatorial-Atlantic tethyan water-mass. These proxies reach maxima along with the abundance of *Bosedinia cf.* sp. 1 & 3 during the deposition of the Hartland Shale Mb. (~95 Ma; t2: **Figure 14**) indicating the marine transgression of equatorial-Atlantic tethyan waters into the KWIS and mirrors the 3[rd] order eustatic Greenhorn Cycle (Kauffman, 1977) reaching the maximum flooding at ~94.7 Ma (see supplemental information). The long term (3[rd] order) trends in water-mass evolution reported here are therefore likely driven by variations in eustasy related to

regional tectonic and/or mantle plume-lithosphere dynamics associated with the emplacement of LIPs (i.e. High Arctic; Caribbean) during this Greenhouse period lacking polar continental ice-sheets.

During peak transgression (~94.7 Ma, t3; **Figure 14**) a significant oceanographic re-organization was marked initially by increased bottom-water current activity resulting in widespread hiatal surfaces across the KWIS and with the rapid onset of more oligotrophic, oxygenated and open marine conditions. These trends are evidenced by increased diversity of dinocyst assemblages dominated by G-cysts; reduction in trace metals abundances, reduced TOC and isorenieratene derivatives (see Sun et al, 2016) as well as increased bioturbation and diverse benthic foraminfera abundances cumulating in the development of the Benthonic/Benthic Oxic Zone (Eicher and Worstell, 1970; Keller and Pardo, 2004).   This environmental shift is interpreted to reflect the rapid southward incursion of a northerly sourced water-mass into the central and southern KWIS and is supported by the equatorial migration of boreal dinocyst taxa (e.g. *Cyclonephelium compactum–membraniphorum* morphological plexus; Eldrett et al. 2014; Van Helmond et al. 2016), occurring during a period of climate cooling with minima (3–5°C cooling) during the PCE (Jarvis et al. 2011; Van Helmond et al. 2014; 2016; Elderbak and Leckie, 2016). Therefore, the increase in abundance and diversity of foraminiferal/molluscan and ammonite assemblages in the KWIS (McNeil and Caldwell, 1981; Kauffman, 1984, 1985; Eicher and Diner, 1985; Elder, 1985; Leckie et al., 1998; Caldwell et al., 1993; Kauffman and Caldwell, 1993, Elderbak and Leckie, 2016) and the lithologic transition from organic-rich mudrocks to limestone dominated facies (Corbett et al. 2014; Lowery et al. 2014) in the latest Cenomanian does not reflect the incursion of tethyan water, but instead the southward flow of a boreal water-mass at this time. Comparable dinocyst assemblages including diversified G-cysts are recorded i) in the Canadian KWIS throughout the Cenomanian and Turonian (e.g. Singh, 1983; Bloch et al. 1999), albeit with higher numbers of boreal P-cyst taxa including *Isabelidinium magnum* and *Eurydinium glomeratum*, and ii) coeval sediments from European shelves (e.g. Foucher, 1980; Jarvis et al. 1988, 2011; Fitzpatrick, 1995; Pearce et al. 2003, 2009; Lignum 2009) which also correspond with the influx of boreal macro-fauna (Jefferies, 1962, 1963; Gale and Christensen, 1996; Voigt et al., 2006; Jarvis et al., 2011).  Evidence for the southward incursion of a northerly sourced water-mass during the latest Cenomanian is recorded in all the studied sections spanning the central and eastern part of the KWIS and therefore does not support a more complex oceanographic system such as that modelled by Slingerland et al. (1996) and Kump and Slingerland (1999). The data presented here indicates that the initial inflow of equatorial-Atlantic tethyan water across the southern gateway is completely replaced by the almost simultaneous southward flow of boreal water (see also Van Helmond et al. 2016), with no evidence for a contemporaneously counter-flowing tethyan water-mass along the eastern margin. However, based on the limited geographic extent of the studied sections these findings are considered tentative, with additional palynological and geochemical investigations on the eastern margin of the KWIS being required to constrain the lateral variability in possible water-mass properties.

Although the migration of boreal dinocyst taxa (namely the *Cyclonephelium compactum–membraniphorum* morphological plexus; van Helmond et al. 2016) during the PCE is not recorded further to the south in the equatorial North Atlantic; a similar shift in palynological assemblages, including a marked decline in

prasinophyte algae, is recorded in the shallower shelf settings on Demerara Rise (ODP Site 1261). Furthermore, at ODP Site 1261, the recorded shift towards a more diverse and open-marine dinocyst assemblage is also associated with an increase in the abundance of organic foraminiferal test linings; re-population by calcareous benthic foraminifera (Friedrich et al., 2006; 2011), which combined with a reduction in redox sensitive trace metals may indicate an improvement in environmental conditions and a reduction on the oxygen minimum zone. However, it should be noted that organic matter concentration remains high. This shift in environmental conditions also corresponds with a sea surface temperature minimum in the lower part of the OAE-2 CIE (up to 4°C; Forster et al. 2007) on the flank of Demerara Rise (ODP Site 1260) and perhaps marks the southernmost influence of a boreal water-mass. However, in Site 1260 *Bosedinia cf.* sp.1 & 3 remains abundant despite the decline in both prasinophyte algae, redox trace metals and the associated increase in benthic foraminifera indicating more complex interaction between water-masses and the oxygen minimum zone along the slope. The southern expression of this boreal influence is therefore limited in duration and extent, mostly affecting shallower water settings. These findings are consistent with Neodymium (Nd) isotope data from Demerara Rise indicative of circulation pattern change in the Atlantic- Tethys at this time (Zheng et al., 2016; Martin et al 2012).

The early Turonian to Coniacian (~94 Ma – 89 Ma, t4-t5: **Figure 14**) interval was characterized by two further proposed incursions of both boreal and equatorial-Atlantic tethyan waters into the KWIS. The first Turonian incursion of boreal water into the southern KWIS (~92.5 Ma) was marked by increased bottom water currents and a 125-210 kyr hiatal interval in Iona-1 (Eldrett et al. 2015a, b). This hiatal interval correlates with that of much greater duration in shallower shelf setting of Innes-1 (> 2 Myrs) and is coeval with the regional middle-lower Turonian hiatus identified throughout the KWIS (e.g. Ewing, 2013). In the central KWIS, at the Pueblo GSSP, the upper part of the lower Turonian and lower part of the middle Turonian substages contain a marked increase in the boreal P-cyst taxa *Isabelidinium magnum* and *Eurydinium glomeratum.* The most notable incursion of equatorial-Atlantic tethyan water in Texas in the Turonian corresponds with the Langtry Mb. of the Eagle Ford Gr. and is characterized by greater abundance of prasinophyte phycomata; greater dinocyst P: G ratio; reduced dinocyst diversity; enrichments in redox sensitive trace metals and enhanced preservation of TOC: all indicative of increased surface water organic matter production and preservation due to partially or episodically anoxic bottom-water or sediment-water conditions. The second main incursion of boreal water into Texas occurs in the Turonian-Coniacian and is marked by the development of chalk facies (e.g. Austin Chalk), diverse dinocyst assemblages and a reduction in redox sensitive trace metals and TOC indicative of oxygenated and oligotrophic marine conditions. Although, the transition from the Eagle Ford Gr., to Austin Chalk is conformable in both Iona-1 and Innes-1, an unconformity is identified on the San Marcos Arch (Lowery et al., 2014). With the exception of the Buda-Eagle Ford unconformity that marks the initial flooding of the southern KWIS by tethyan waters; the other hiatal surfaces identified throughout the KWIS appear to be associated with increased bottom-water current activity linked to regional climatic and/or tectonic subsidence-induced incursions of boreal water that ventilated the seaway rather than solely by global eustatic drivers that are difficult to reconcile with a greenhouse world without significant polar ice. The development of coeval hiatal intervals are also recorded near the top and base of the OAE-2 CIE in the Bonarelli interval, Italy, where vigorous bottom

currents were proposed to be induced by warm and dense saline deep waters that originated on tropical shelves in the Tethys and/or proto-Atlantic Ocean (Gambacorta et al. 2016). Although our contribution proposes an alternative mechanism for invigorated circulation, it further supports the suggestion by Gambacorta et al. (2016) that enhanced current activity and associated oceanographic circulation was much more dynamic than previously thought during times of Greenhouse climates when conditions were thought to be more equitable.

### 4.2.3.    Global trace metal draw-down during OAE-2

During the Cenomanian, sediments influenced by equatorial-Atlantic tethyan waters record redox sensitive trace metals that are enriched compared to average shale, in particular Molybdenum (Mo) which requires free $H_2S$ for authigenic sedimentary enrichment (Helz et al. 1996) and is interpreted as reflecting anoxic to euxinic depositional conditions favouring organic matter preservation (see Algeo and Lyons, 2006).  The relative enrichment of Mo compared to the other redox elements such as U and V in the pre-OAE-2 lower Eagle Ford Gr., suggests an active cycling of Mn–Fe particulates (especially Mn-oxyhydroxides) between the water column and sediment water interface characteristic of a "particulate shuttle" (Algeo and Tribovillard, 2009; Tribovillard et al. 2012; **Figure 15a**). The presence of aryl isoprenoids in this interval from the Iona-1 core also supports at least temporary and partial water column (photic zone) euxinia at this time (Sun et al. 2016).

During the onset of OAE-2 the recorded depletion of sedimentary redox sensitive trace metals has been proposed to reflect the draw-down of the global trace metal sea-water inventory due to sequestration in sediments under expanded anoxic/euxinic conditions (Hetzel et al. 2009; Dickson et al. 2016; Goldberg et al. 2016). These interpretations are based primarily on the exceptionally low Mo/TOC gradients compared to those documented in modern-day anoxic silled marine basins (Algeo and Lyons, 2006), whereby removal of aqueous Mo concentrations under anoxic/euxinic conditions to the sediment is in excess to resupply by inter-basinal transfer of water masses. This contribution also documents exceptionally low Mo/TOC gradients for sediments spanning the OAE-2 interval from Demerara Rise and SW Texas (**Figure 15b**; also see Hetzel et al. 2009; Eldrett et al. 2014 Fig. DR3), but also from those recovered from the upper Eagle Ford Gr., and from the Dakota Sandstone interval from the Portland-1 core (supplemental datafile).

However, there are three main challenges to the application of Mo/TOC gradients to infer global trace metal inventory drawdown during OAE-2.  Firstly, during the OAE-2 interval in the KWIS, Demerara Rise and Kerguelen Plateau, the weak positive co-variance between Mo and U indicates deposition along the unrestricted open marine trend (**Figure 15a**; also Eldrett et al. 2014); an interpretation supported by the palynological assemblages presented here. Therefore, during the OAE-2 interval, these localities were not subject to significant hydrographic restriction (e.g., silled basins), being deposited instead along continental margins. Thus the application of modern-day Mo/TOC gradients to infer hydrographic restriction and trace metal drawdown for OAE-2 may not be valid and requires additional study (see discussion in Algeo and Lyons, 2006).

Secondly, as Mo is mostly incorporated into organic matter, we propose that the various contributions of different types of organic matter would affect Mo/TOC gradients. Whereas sulfurized organic matter is known to

enhance Mo uptake by the sediment (Tribovillard et al. 2004), the impact of variable contributions of labile versus refractory organic matter is not well constrained. The palynological analyses presented here demonstrate increased contribution of refractory terrigenous organic matter (>T:M ratio) in the samples with low Mo/TOC values from the KWIS and Demerara Rise (**Figure 15b**). We cannot determine the impact of this observation as the T:M ratio reflects a relatively small proportion of the total refractory organic matter, however further investigation is warranted into the variable origin of the more dominant and relatively unknown component, namely AOM; and whether Mo is preferentially incorporated within different organic matter components.

Thirdly; in all the sections presented here the depletion in redox sensitive trace metals during OAE-2 is either associated with oxygenated depositional conditions, or an improvement in redox state. Therefore, the recorded depletion in redox sensitive trace metals, in particular Mo may instead record a genuine environmental response to benthic oxygenation preventing authigenic sedimentary enrichments as under oxic conditions, Mo is present in seawater as the stable and largely unreactive molybdate oxyanion. These observations are interpreted to be related to the equatorial migration of boreal water-masses promoting water column de-stratification and partial ventilation/re-oxygenation of the surface - deep waters.

These challenges were partly addressed by Dickson et al. (2016) who proposed that the recorded low Mo/TOC gradients from Site 1138, Kerguelen Plateau reflected the widespread and global nature of trace-metal depletion during OAE-2 by demonstrating that i) localized oxygenation was also not responsible due to Mo isotope compositions and ii) organic matter type was not a control due to the low abundance of terrigenous material. Palynological investigation of these sediments (this study) confirms the low abundance of terrigenous material at Site 1138. In addition, the dinocyst assemblage recovered from this site is typical of a well-oxygenated boreal equivalent (austral) water-mass with high diversity and abundance of G-cysts, notably *Cyclonephelium compactum- membraniphorum* dinocyst plexus, and outer neritic to open marine taxa such as *Spiniferites ramosus* and *Pterodinium cingulatum*. These findings would suggest deposition under the influence of an oxygenated and open marine/hydrographically unrestricted austral (Southern Hemisphere) surface water-mass on the Kerguelen Plateau; consistent with the relatively low $\delta^{98/95}$Mo values and Mo-U co-variance trends reflecting oxic-suboxic depositional conditions; with at most sulphidic pore-waters (Dickson et al. 2016). Furthermore, conditions at the sediment-water interface were sufficient for the presence of a low diversity, high carbon-flux benthic foraminifera biofacies (Holbourn and Kuhnt, 2002) as also indicated by the rare occurrence of organic linings of benthic foraminifera throughout the OAE-2 interval at Site 1138 (this study). Therefore, in addition to trace metal drawdown as an explanation to reconcile low Mo/TOC gradients and hydrologically unrestricted regime, it is possible that even in the organic-rich laminated sedimentary intervals (i.e. Sites 1138 and 1260) a slight increase in oxic conditions related to the influence of oxygenated high-latitude austral-boreal water-masses, may result in aqueous $H_2S$ concentrations at the sediment- water interface being below the critical threshold for conversion of molybdate to thiomolybdate (Helz et al. 1996), resulting in the observed depletion in Mo sedimentary concentrations and low Mo/TOC gradients during OAE-2. This interpretation is also supported by the occurrence of benthic foraminifera within the OAE-2 interval in both organic rich and organic lean sedimentary sections from Demerara Rise (Friedrich et al. 2006, 2011); Kerguelen Plateau (Holbourn and Kuhnt,

2002); Texas (Lowery et al. 2014; Dodsworth, 2016; this study) and central KWIS (e.g. Eicher and Worstell, 1970; Keller and Pardo, 2004; Elderbeck and Leckie, 2016 and references therein).

### 5. Conclusions

This integrated palynological and geochemical study using a multidisciplinary approach has provided insights into depositional environments of the Cenomanian-Turonian Eagle Ford Gr. and bounding formations of the Buda Limestone and Austin Chalk from southwest Texas, USA. The study spans a physiographic transect across the Comanchean Shelf from the shallow setting of the San Marcos Arch (AU2); through the main shelf (Innes-1), the slope (Iona-1) to the intra-shelf basin (well X). Furthermore, comparison of the paleoenvironmental trends recorded in the Cenomanian – Coniacian section from SW Texas to those recorded further north within the central KWIS (USGS Portland-1) and to the south in the equatorial North Atlantic (ODP sites 1260, 1261, Demerara Rise) and Kerguelen Plateau (ODP Site 1138) allows the interpretation of the evolution of various water-masses along this north-south transect. The main findings are:

1.  High latitude boreal and austral (Northern and Southern Hemisphere) and equatorial Atlantic tethyan water-masses can be distinguished based on their distinct palynological assemblages and geochemical signatures.

2.  The northward flow of a suboxic-anoxic equatorial Atlantic tethyan water-mass into the Western Interior Seaway occurred during the early-middle Cenomanian; followed by a major re-organisation of the oceanographic regime during the latest Cenomanian- Turonian as a full connection of the Western Interior Seaway with a northerly- boreal water-mass was established during peak transgression. This oceanographic change promoted de-stratification of the water column and improved oxygenation throughout the KWIS and as far south as the Demerara Rise.

3.  These long term trends in water-mass evolution are tentatively linked to third order eustatic transgression-regression cycles driven by regional Cordilleran tectonic and/or mantle plume-lithosphere dynamics associated with the emplacement of LIPs during this time as well as shorter term variations in climate (i.e. Plenus Cold Event).

4.  Low Mo/TOC ratios in the equatorial North Atlantic in comparison to other oceanic basins during the onset of OAE-2 argue for partial restriction and draw-down of global trace metal sea-water inventories. However, this study demonstrates that the recorded decline in redox-sensitive trace metals during the onset of OAE-2 likely reflect either a genuine oxygenation event related to open water-mass exchange at this time and/or is further complicated by variable contribution of organic matter from different sources (e.g. refractory/terrigenous material) that requires further evaluation.

### 6. Acknowledgements

This paper used data generated on sediments recovered and curated by the International Ocean Discovery Program. The authors thank Malcolm Jones and PLS Ltd for palynological preparations; Mark Phipps and John Gregory for micropalaoentological analyses at Petrostrat Ltd. In addition, we thank all our Shell colleagues who contributed, in particular Iain Prince for technical discussion and internal review and Shell Exploration and

Production Inc., for permission to publish.

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

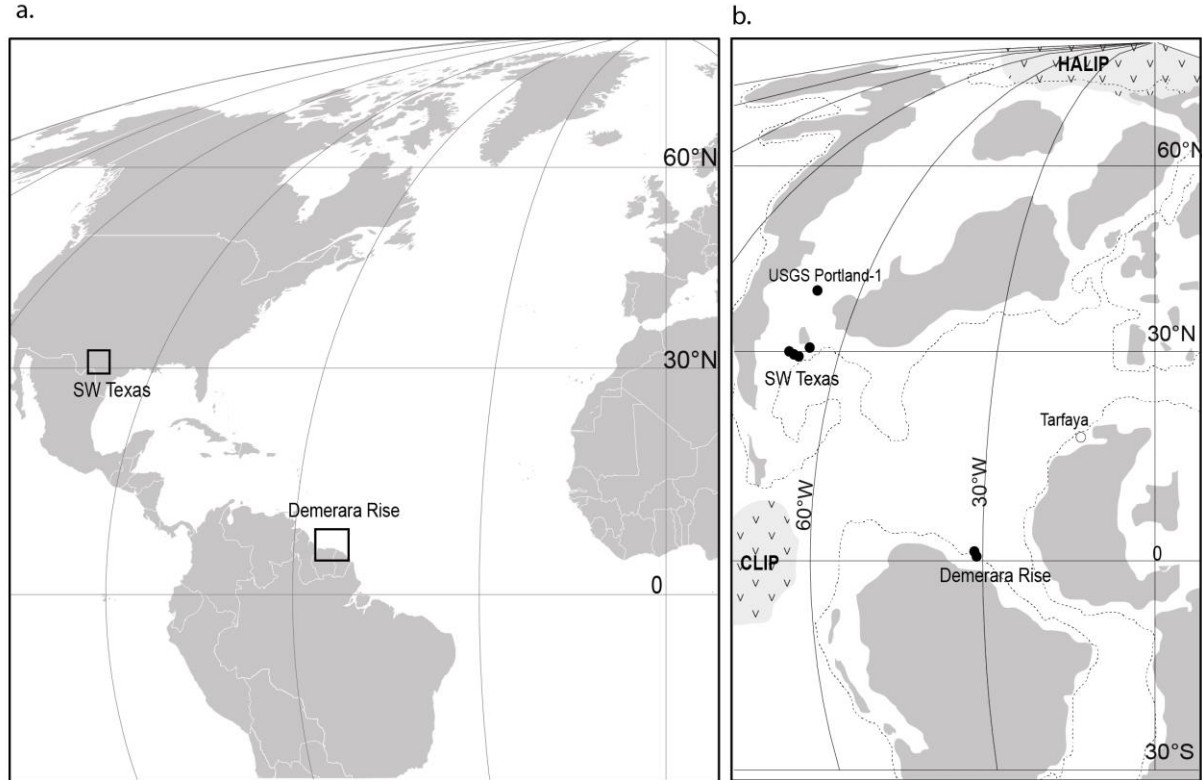

**Figure 1:** Site Locations. a. Present-day position of the study areas; b. Turonian paleogeographic reconstruction with site locations; gray shaded area= landmass; dotted line = paleo-shelf; CLIP =Caribbean Large Igneous Province; HALIP = High Arctic Large Igneous Province. Boxes show study area as presented in Fig. 2

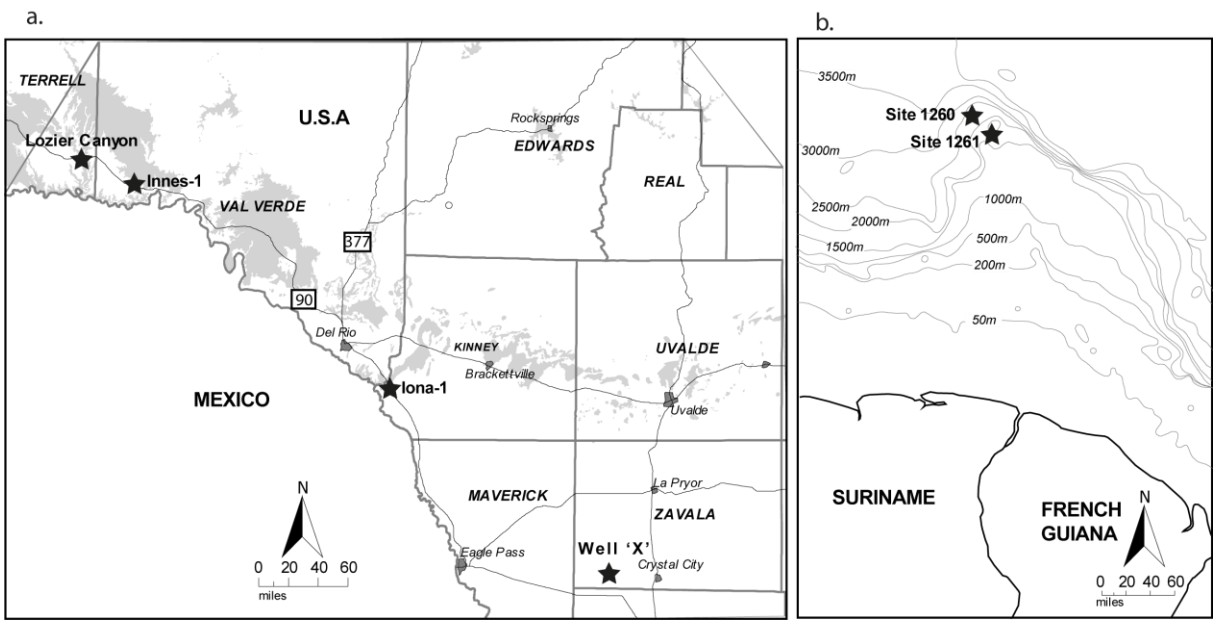

**Figure 2:** Site Locations. **a.** SW Texas map showing core locations (black stars); gray shading = Eagle Ford Gr., outcrop belt; **b.** Demerara Rise site locations.

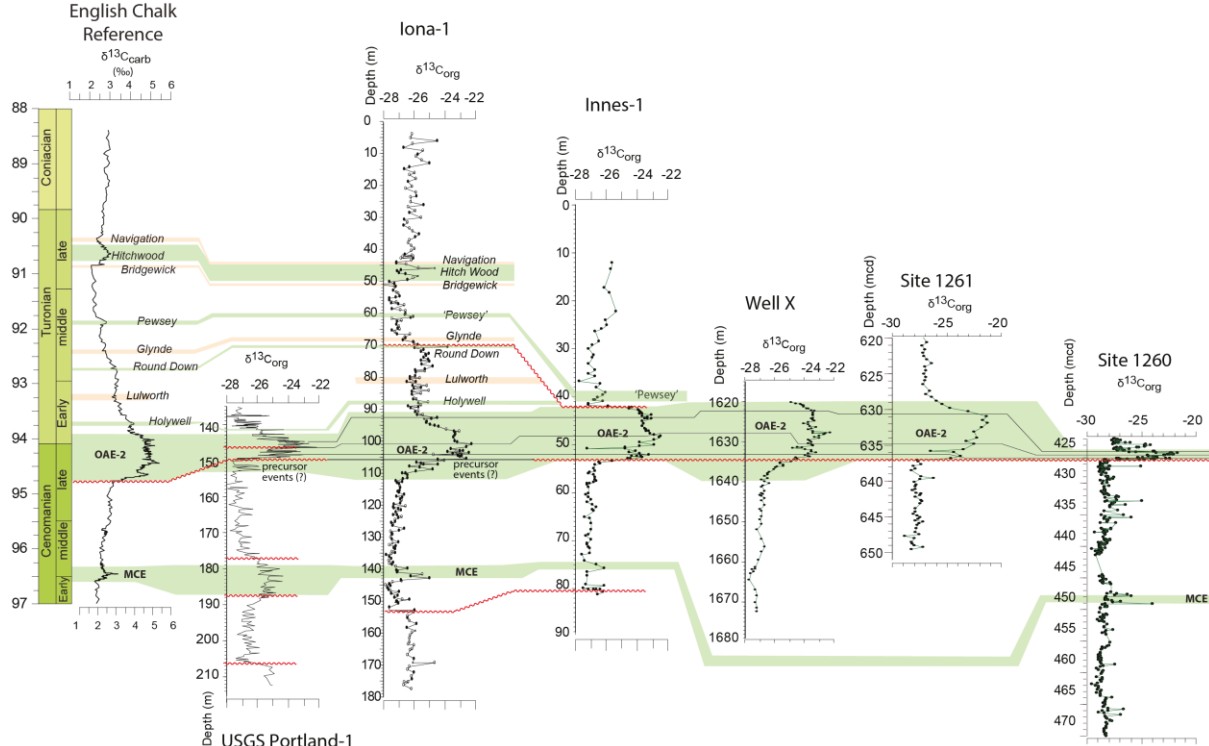

**Figure 3:** Isotope data and correlation between studied sites and the English Chalk Reference section. Data sources: $\delta^{13}C$ for English Chalk (Jarvis et al. 2006); USGS Portland-1 (Joo and Sageman, 2014; Eldrett et al. 2014; Duvivier et al. 2014; Eldrett et al. 2015a, this study) ODP sites 1260-1261 (Erbacher et al. 2005; Friedrich et al. 2008) Note variable depth scales. Wavy horizontal red lines represent hiatal surfaces. Green shading = positive $\delta^{13}C$ isotope events; Orange shading = negative $\delta^{13}C$ isotope events. Isotopic events nomenclature follows that of Jarvis et al. (2006); with the exception of precursor events (after Eldrett et al 2014); which are now though to occur within the OAE-2 CIE; but in the section not recorded at Portland-1 core or the Pueblo GSSP due to the hiatal surface at the base of the Bridge Creek Limestone.

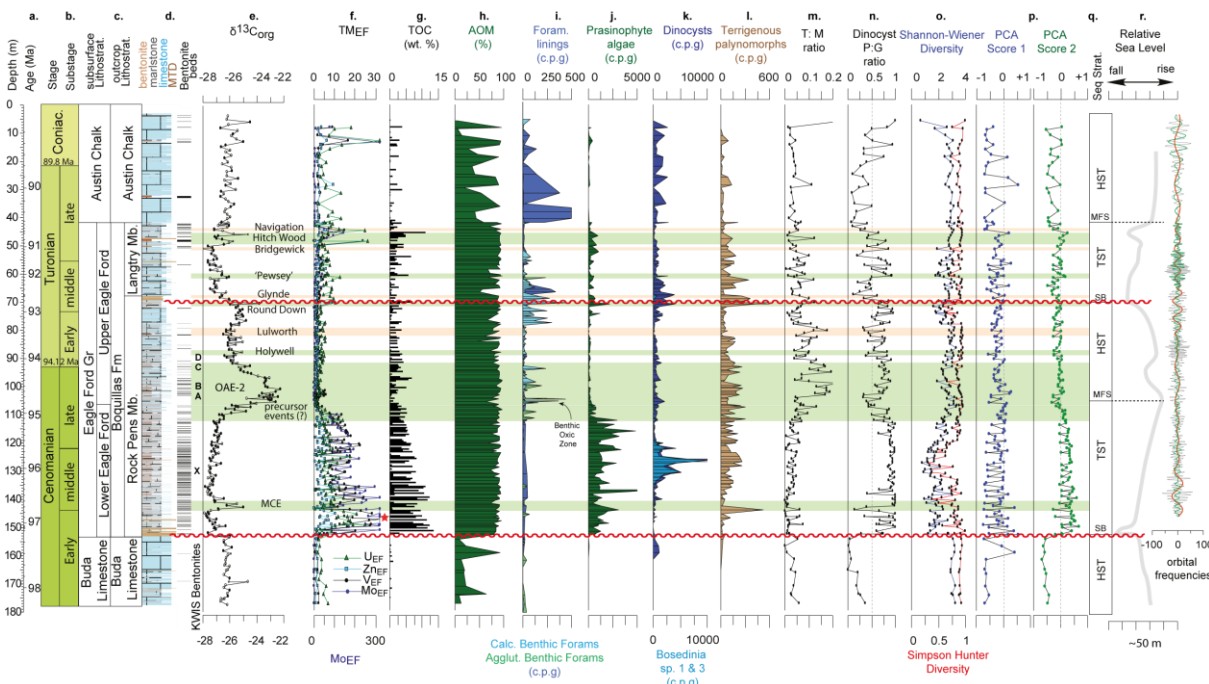

**Figure 4:** Iona-1 core data. a. depth/age; b. chronostratigraphy; c. lithostratigraphy; d. lithology; e. $\delta^{13}C_{org}$; f. redox sensitive trace metal enrichments ($TM_{EF}$); g. TOC; h. AOM; i. foraminiferal test linings; micropalaeontological abundance data as counts per gram (c.p.g) for both calcareous (Calc.) and agglutinated (Agglut.) benthic foraminifera; j. prasinophyte algae; k.

dinocysts, *Bosedinia cf.* sp. 1 & 3; l. Terrigenous palynomorphs; m. T:M ratio; n. dinocyst P:G ratio; o. Shannon-Wiener and Simpson-Hunter dinocyst diversity; p. Principle Component results; eigen score 1, eigen score 2; q. Sequence stratigraphic interpretations (see supplemental information); r. inferred relative sea level history (after Minisini et al. in review), with Milankovitch orbital frequencies (after Eldrett et al. 2015a, b).

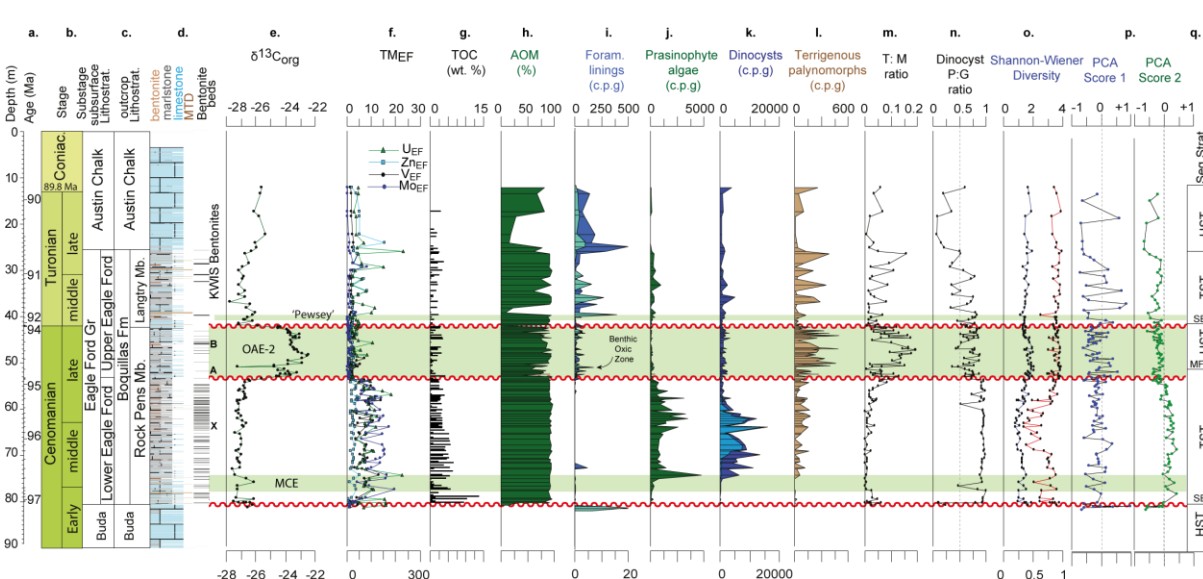

**Figure 5:** Innes-1 core data. Legend same as Figure 4. Lithology after Minisini et al. (in review).

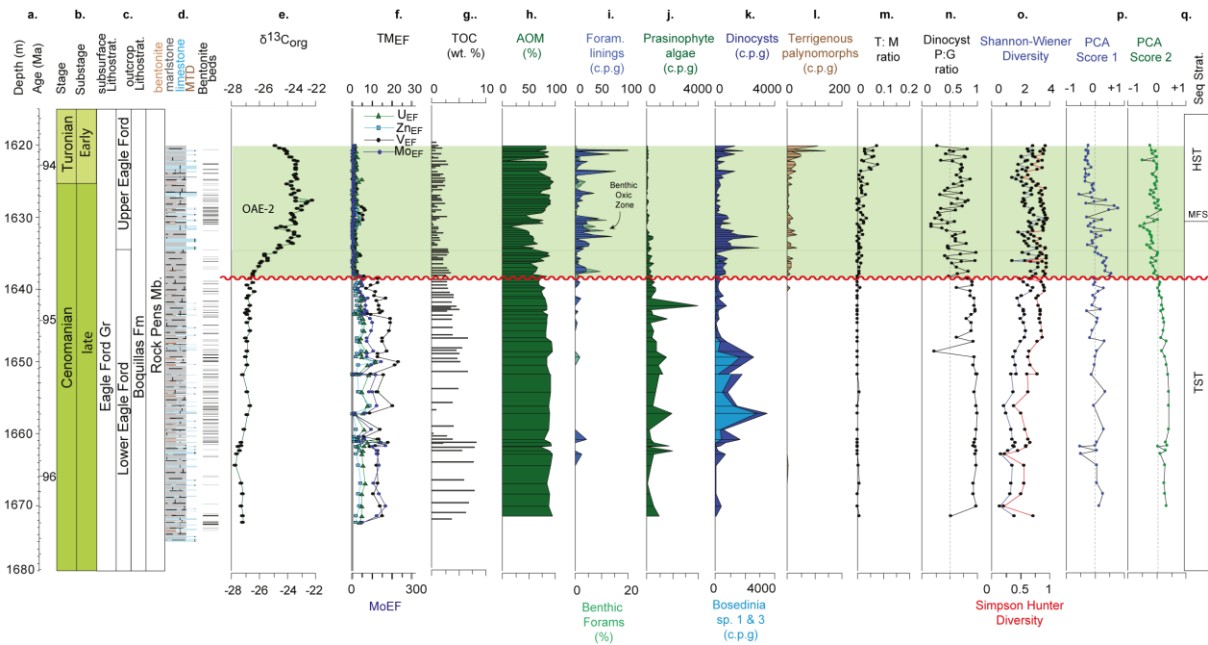

10     **Figure 6:** Well 'X' core data. Legend same as Figure 4.

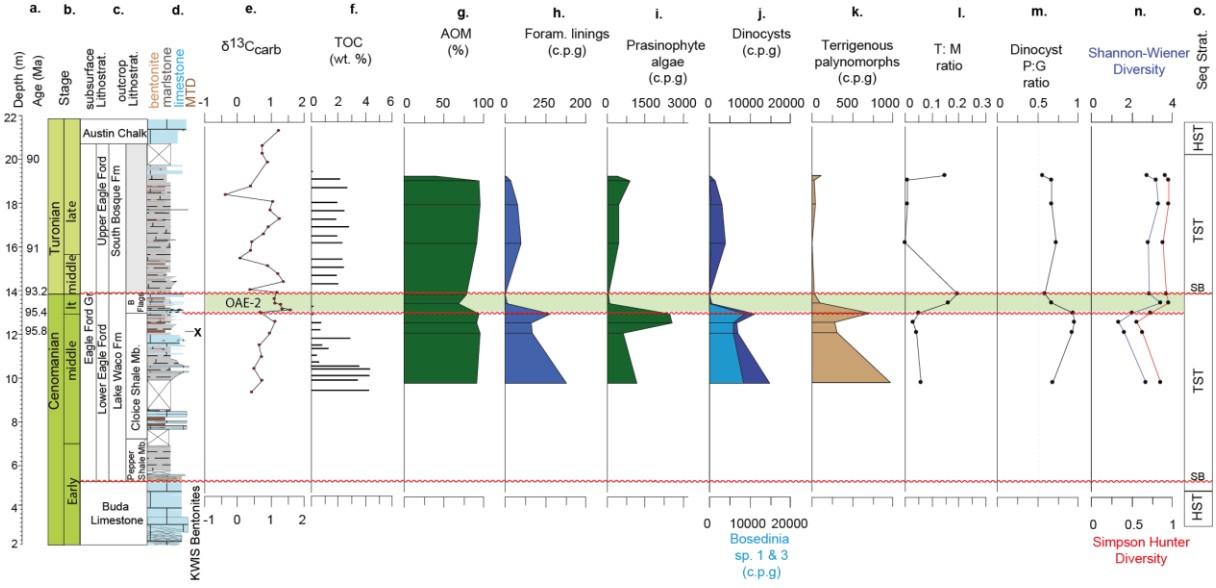

**Figure 7:** Bouldin Creek outcrop (AU-2) data. Legend same as Figure 4. Lithology after Minisini et al. (in review). $\delta^{13}C_{org}$ and TOC (wt. %) from Lowery et al. (2014)

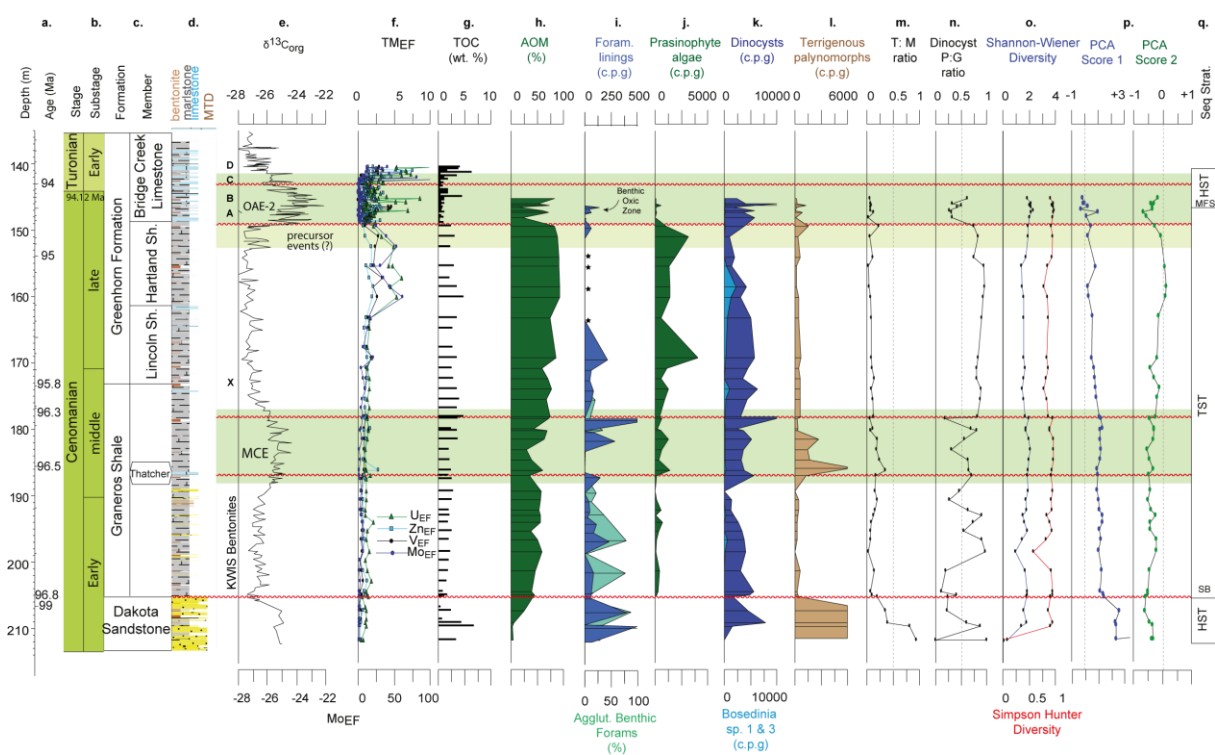

**Figure 8:** USGS Portland-1 core data. Legend same as Figure 4. $\delta^{13}C_{org}$ data from Joo and Sageman (2014); Eldrett et al. (2014); Duvivier et al. (2014); Eldrett et al. (2015a), this study.

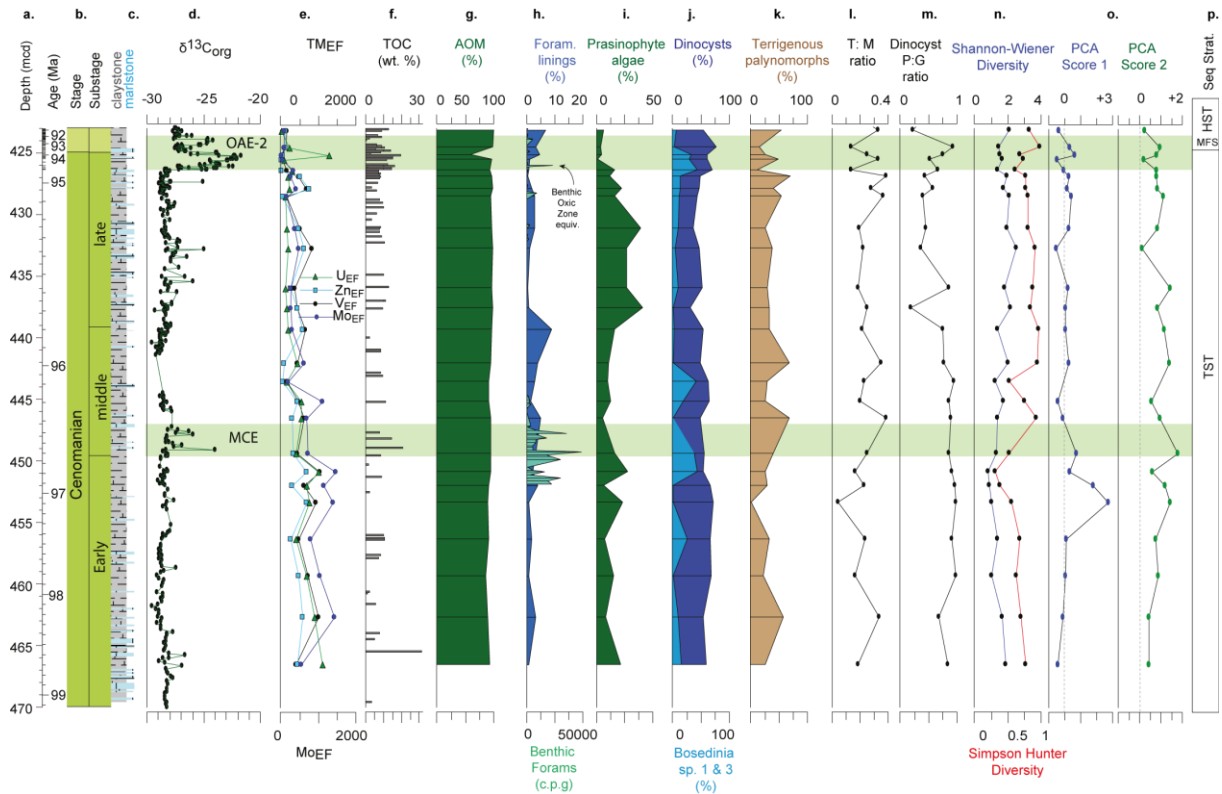

**Figure 9:** ODP Site 1260 data. Legend same as Figure 4. δ¹³C_org data from Erbacher et al. (2005); Friedrich et al. (2008)**.** Benthic foraminiferal abundance data from Friedrich et al. (2006).

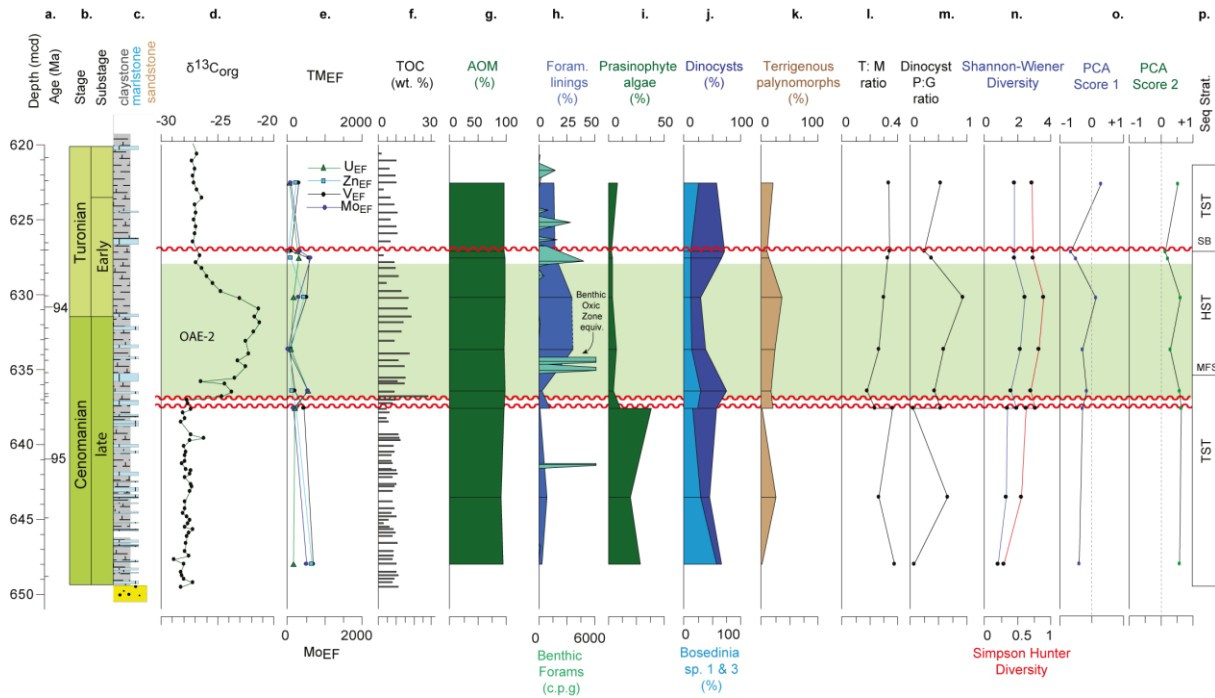

**Figure 10:** ODP Site 1261 data. Legend same as Figure 4. δ¹³C_org data from Erbacher et al. (2005) and Friedrich et al. (2008). Benthic foraminiferal abundance data from Friedrich et al. (2006).

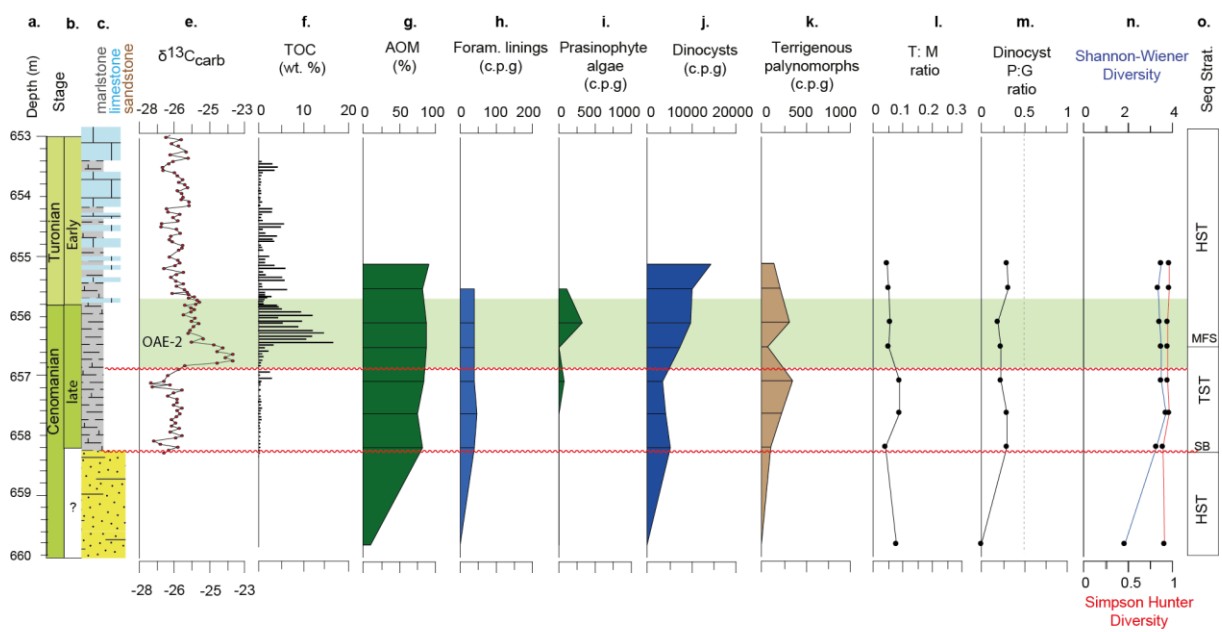

**Figure 11:** ODP Site 1138 data. Legend same as Figure 4. $\delta^{13}C_{org}$ and TOC data from Dickson et al. (2016).

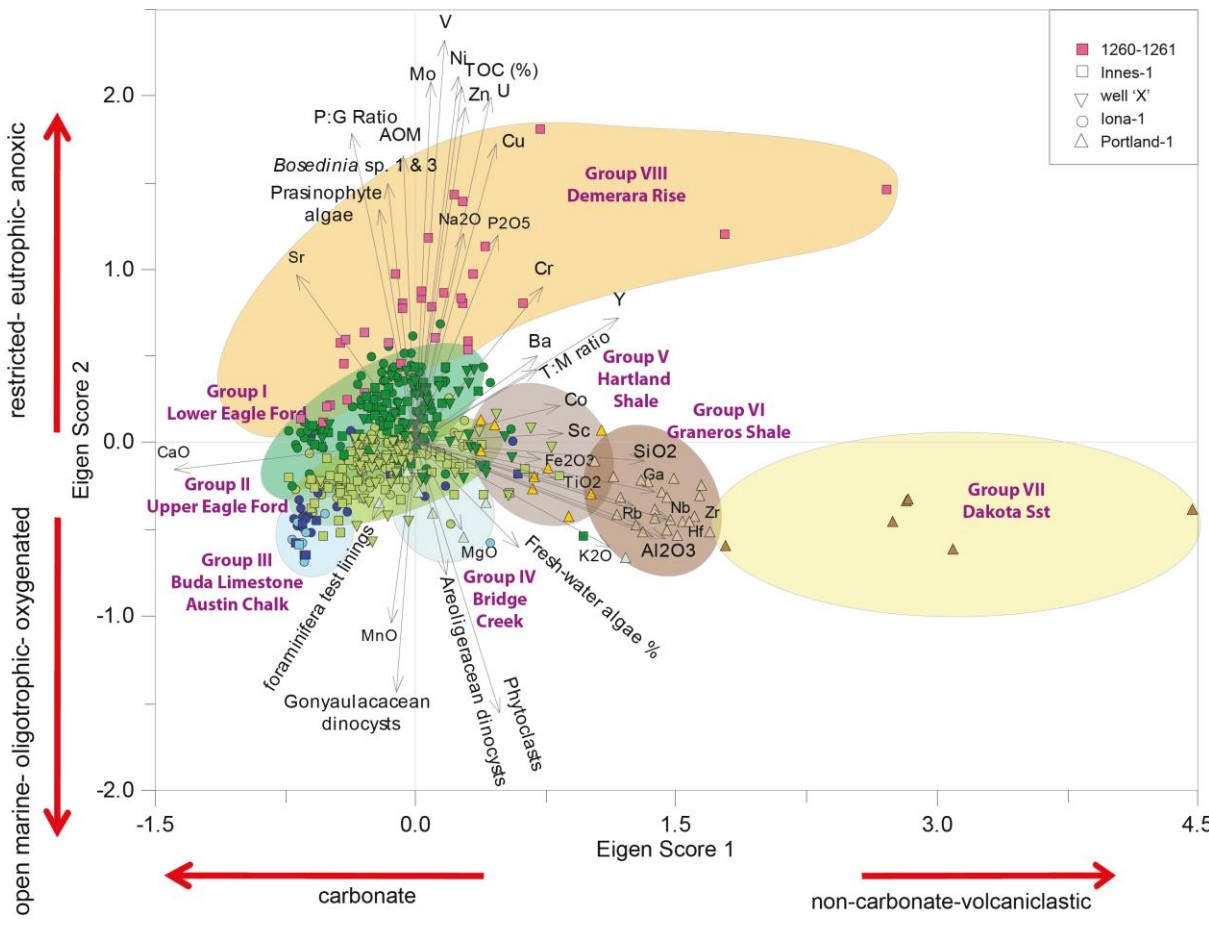

5 **Figure 12:** PCA results showing principle axis/eigen scores; selected environmental variables. Samples plotted as a function of site location (see insert) and lithostratigraphic interval (colours; groups in figure).

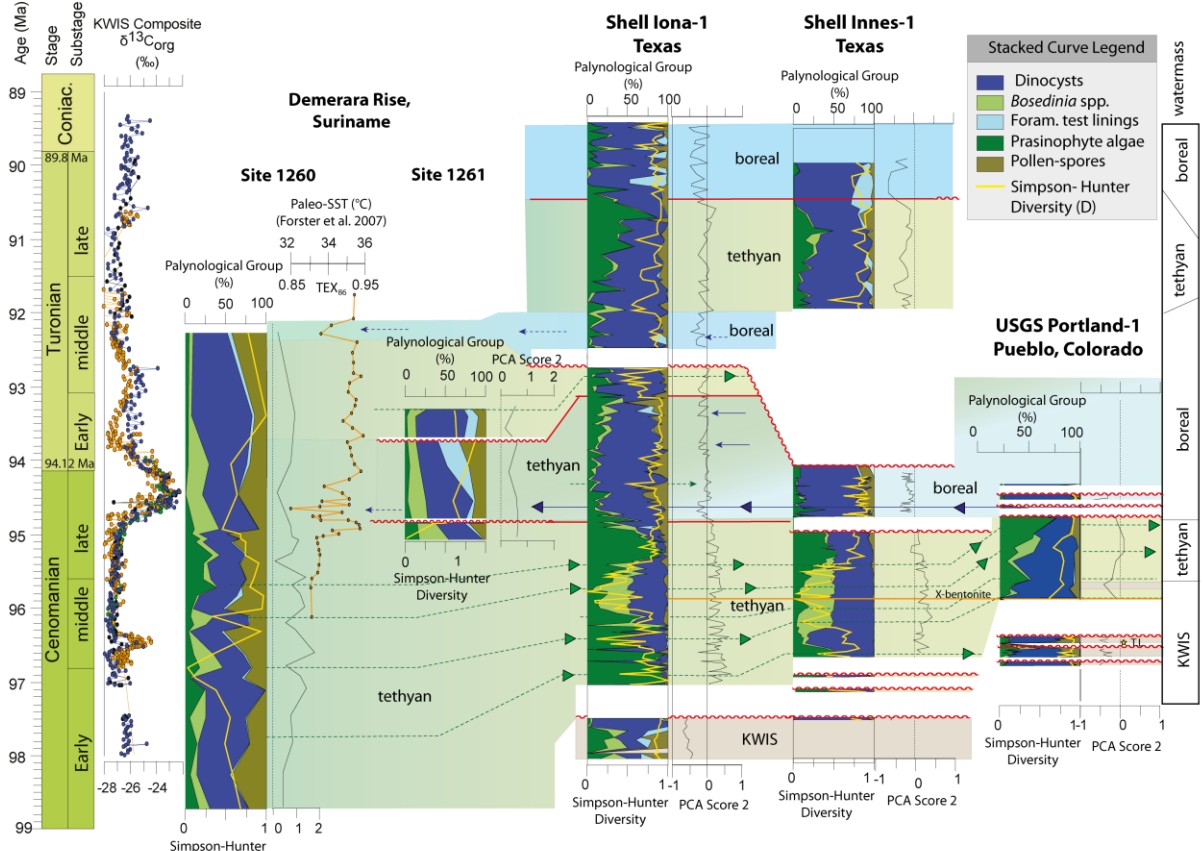

**Figure 13:** Cenomanian-Coniacian chronostratigraphic correlation of interpreted water-masses along a South-North transect from the equatorial western Atlantic to the central Cretaceous Western Interior Seaway (KWIS). The KWIS composite $\delta^{13}C_{org}$ after Joo and Sageman (2014) and including data from Duvivier et al. (2014), Eldrett et al. (2015) for USGS Portland-1 (orange), plus data from Iona-1 (blue); Innes-1 (black) and well 'X' (green) cores calibrated using the age model of Eldrett et al. (2015a). Stacked curves showing principal palynological components (legend on figure). Paleo-SST and TEX$_{86}$ record from ODP Site 1260 from Forster et al. (2007). Colour shading: green = tethyan sourced water from the equatorial Atlantic; blue = northerly boreal watermass; brown = locally sourced KWIS watermass. Green dashed arrows indicate tethyan incursions; blue arrows indicate boreal influence. Principal Component Score (PCA) axis 2; positive indicative of eutrophic/anoxic waters; negative indicative of oxygenated oligotrophic waters (see Figure 12).

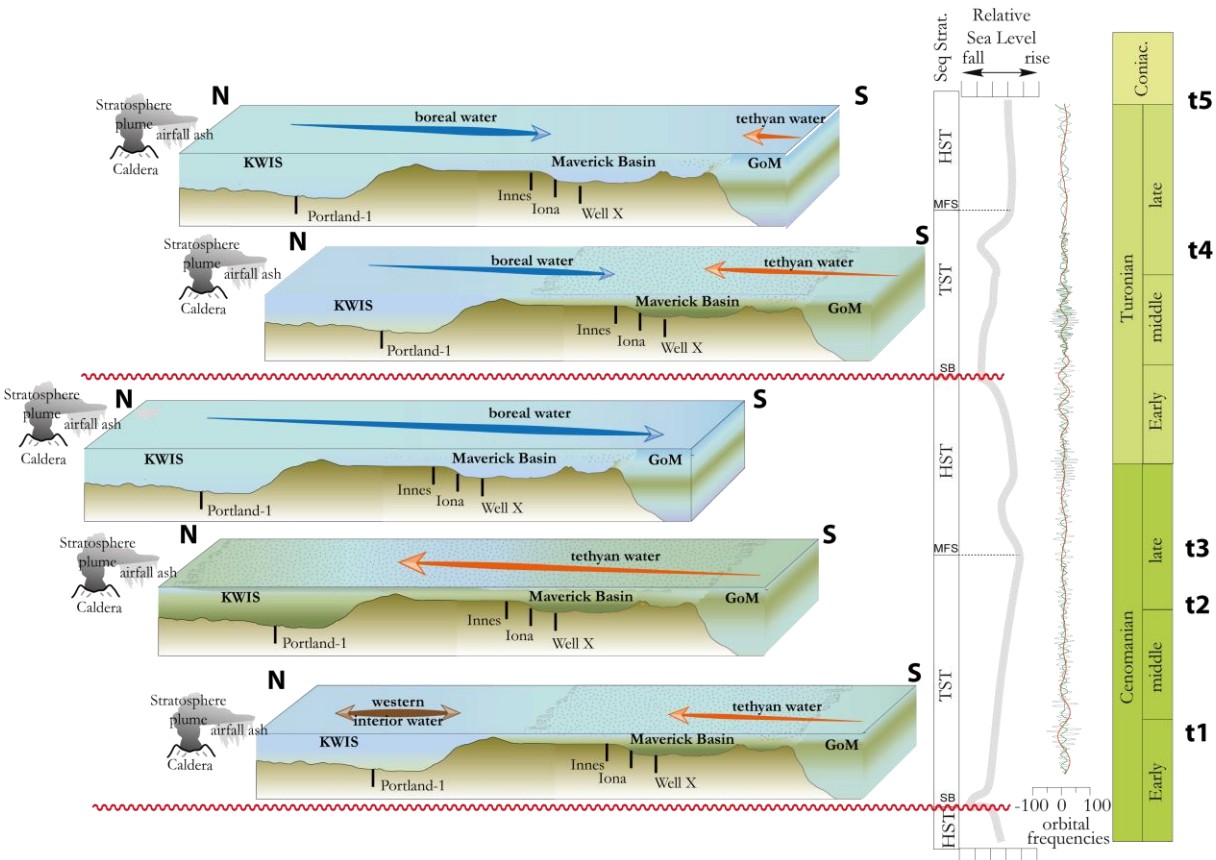

**Figure 14.** Regional water-mass depositional model of the KWIS. Time-slices (t1-t5) showing the northward flow of eutrophic/anoxic tethyan equatorial Atlantic water into the Western Interior Seaway from and southward flow of a boreal watermass. Milankovitch orbital frequencies of eccentricity, obliquity and precession from the Iona-1 core (Eldrett et al. 2015a) resulting in shorter-term (<0.1 Ma) development of limestone-marlstone couplets (Eldrett et al. 2015b). GoM = Gulf of Mexico. Sea level curve after Minisini et al., (in review).

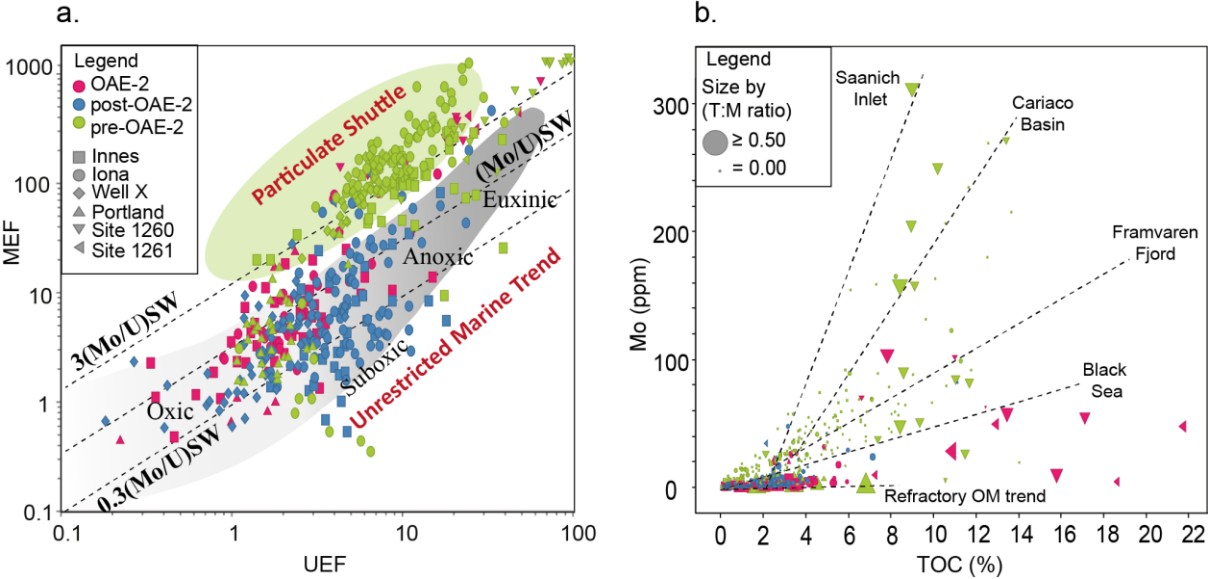

**Figure 15. a,** Logarithmic plot of Mo:U covariance for the studied sections. Diagonal dashed lines represent multiples (0.3, 1, and 3) of Mo:U ratio (3) of present-day seawater (SW). Values greater than x3 SW are generally associated with the particulate shuttle; whereas those trending with modern day sea water reflect an unrestricted marine trend with a weak positive Mo:U covariance. **b,** Mo-TOC covariance data for studied sections compared to modern anoxic silled-basin

environments (after Tribovillard et al. 2012). Regression of the modern datasets are shown as solid lines with MO/TOC regression slopes displayed (Tribovillard et al. 2012). The proposed refractory organic matter trend added as dashed line.