# Peer review of "Water-mass Evolution in the Cretaceous Western"

_Climate of the Past, 2016_

## Referee Comment (RC1) · Anonymous Referee #1 · 9 Jan 2017

The study by Eldrett and colleagues tries to understand and constrain the nature and timing of water-mass evolution in the southern gateway to the Cretaceous Western Interior Seaway (KWIS) and define the associated paleoenvironmental and paleoclimatic processes. This article presents detailed palynological and geochemical analyses from Cenomanian-Conacian interval from southwest Texas,USA. They also analysed materials from the central KWIS, to the south in the Tropical Atlantic and Southern Ocean in order to do correlations and reconstruct N-S water mass circulation. This work proposes another model of ocean circulation in the KWIS to explain the presence of normal oxygenation conditions during CIE (OAE-2) in this zone. It shows, inter alia, a link between sea level variations and N-S oceanic circulation reorganization. Finally, authors

discuss and try to explain the global trace metal drawdown during OAE-2 This is an interesting article with many new data. It provides opportunities to discuss the impact of N-S water mass exchanges on regional paleoenvironmental changes during a period with a predominant latitudinal water mass circulation. However, many portions/sections should be clarified or developed (- discussion about trace-metal significance (detrital versus authigenic origin); - discussion about sequential stratigraphy; - discussion about tethyan water mass in the western part of the Central Atlantic ocean. . .) and some paragraphs must be organised differently in order to help the reader (result and interpretation presentation with the same stratigraphic subdivision that is chronostratigraphic subdivision).

• p.4 ligne 36 : change "paleonenvironmental" by paleoenvironmental

• p.5 ligne 25 : "previously published organic and inorganic geochemical analyses" put bibliographic references

• 2. Material and Method

In the supplementary data there is a lack of discussion on the authigenic origin of the trace metals used. To validate their types of source, the major and trace element abundances have to cross-correlated with Al abundances (indicators of detrital influx). See for example Lebedel et al., 2013 (Pal, Pal., Pal.)

You can also distinguish between redox proxies, Mo, V, U (Calvert and Pedersen, 1993) and palaeoproductivity proxies, Zn, Ni (Hatch and Leventhal, 1992).

Finally, there is no Measurement accuracy.

• 3.1 Organic carbon-isotope Stratigraphy

In this section I think it is important to confront geochemical correlations with biostratigraphic correlations. This section seems to suggest that the identification of the Middle Cenomanian event and the Cenomanian-Turonian CIE is not based on age, but only on the magnitude of $\delta13$ Corg positive excursion.

• 3.2 Geochemistry p.6

It is necessary to re-organize this section. Use chronostratigraphic subdivision instead of lithostratigraphic subdivision. It lacks a description of the diverse sedimentary series that is the description of the main facies because the paleoenvironmental perturbations are recorded in the litho and biofacies. Are there cherts in these series?

• 3.3 Palynology p.7

In order to help the reader, it's better to present the meaning of the different parameters studied and presented in figures 4-10 (for example T/M ratio, Dinocyst P/G ratio, Shannon-Wiener diversity and Simpson Hunter Diversity). This information should not only appear in the supplementary data.

• 4.1 Âń Principal Component Analyses Âż (p. 8), Use chronostratigraphic subdivision instead of lithostratigraphic subdivision (same subdivision as geochemistry section).

• p.8 line 19 "Gallium (Ga)-Al2O3 [kaolinite]" I don't understand the direct link between Al2O3 and Kaolinite. Illite also contains Al2O3

• p. 8 line 27 Âń redox sensitive trace metal concentrations Âż. Zn and Ni are also paleoproductivity proxies

• p.8 line 35 change "paleonenvironmental" by paleoenvironmental

• p. 9 lines 19- 23 "In addition, it is interesting to note that although phytoclasts plot negatively along eigen axis 2 and may represent a reduced masking effect of AOM during oxygenated conditions (Tyson, 1995); they also plot positively along Eigen score 1 (noncarbonate/ volcaniclastic trend) alongside freshwater algae and Areoligeracean dinocysts suggestive of a more nearshore environment (Brinkhuis and Zachariasse 1988; Harker et al. 1990; Li and Habib 1996)". This last interpretation must be confirmed by the calculation of a correlation coefficient which, in my opinion, will not show a correlation. Here, you do only a suggestion but not a real interpretation because the

position of the phytoclasts is not at all correlated to the axis 1

• p. 9 line 25 change "Figures 4-11" by Figures 4-6, 8-10

• p.9 "Paleoenvironment Interpretation"

Use the same chronostratigraphic subdivision as geochemistry section (3.2).

If the sequence stratigraphy interpretation was not published before it's important to explain it.

• p.11, line 16-19 "The sporomorph assemblages during OAE-2 mainly record a relative increase in gymnosperms, in particular during the PCE interval, and thus any increase in T:M ratio may reflect transition from mega-thermal to meso-thermal vegetation (perhaps also reflecting increased pollen production by wind dispersed gymnosperms) in response to climate cooling episode rather than increased hydrologic cycle". Ti/Al can be used as a proxy for eolian versus fluvial input. What does this proxy show?

• P.11 "Regional water-mass evolution"

"In this study we infer three main water-mass properties: i) a restricted suboxic-anoxic marine water-mass characterized by low diversity dinocyst assemblages interpreted to represent a tethyan source; ii) an unrestricted/open marine oxygenated water-mass characterized by high diversity dinocyst assemblages interpreted to represent a boreal source and iii) a partially restricted dysoxic water-mass interpreted to represent a more local central KWIS source."

It's too direct, you have to explain!! Why do you talk about a tethyan source and no an Atlantic-tethyan source? You specify Âń a restricted suboxic-anoxic marine water-mass of tethyan source" before and after CIE. However Tethyan marine waters are generally well-oxygenated during these time intervals which are not the case of the Atlantic marine waters (see ref Monteiro et al., 2012 Paleoceanography for example. . .) ". It would be better to write water mass of Atlantic-Tethyan source.

Section organization: use the same chronostratigraphic subdivision than previously.

• p. 12 lines 8-12 "Lower Cenomanian sediments from the Equatorial Atlantic (ODP Site 1260) are interpreted as being deposited in a stratified suboxic-anoxic marine environment as indicated by laminated organic rich mudrock deposition and positive PCA-2 scores. This interpretation is consistent with a southern tethyan water-mass and a circulation controlled nutrient trap fuelling surface water productivity and anoxic depositional environment (Jiménez Berrocoso et al. 2010; Trabucho-Alexandre, 2010)".

I don't understand why it's consistent with a southern tethyan water-mass. According Trabucho-Alexandre et al., 2010, this zone is the seat of upwelling of deep waters coming from the Pacific, no southern tethyan water-mass is mentioned by these authors.

• p.12 line 17 : "with mixed dinocyst assemblages". List of the genera

• p.12, line 28 "In the Portland-1 core there is a clear shift from agglutinated to calcareous benthic foraminifera near the top of the MCE interval suggestive of a tethyan influence"

why? Is there no agglutinated and calcareous benthic foraminifer in shallow water environment in the Atlantic and Pacific oceans?

• p.13, lines 35-39. "Furthermore, at ODP Site 1261, the recorded shift towards a more diverse and open-marine dinocyst assemblage is also associated with an increase in the abundance of organic foraminiferal test linings; re-population by calcareous benthic foraminifera (Friedrich et al., 2011), which combined with a reduction in redox sensitive trace metals is indicative of an improvement in environmental conditions and a reduction on the oxygen minimum zone".

Warning ! There are few samples analysed; 4 samples in 10 m, it's little. This interpretation does not seem really justified because the organic matter concentration is very high.

• p.15 paragraph Âń Global trace metal draw-down during OAE-2" line 3, "During

the Cenomanian, sediments that have been influenced by tethyan waters"

---

## Referee Comment (RC2) · Anonymous Referee #2 · 24 Mar 2017

The study by Eldrett et al is a very nice contribution about the mid Cretaceous Western interior seaway and how WIS sediments are influenced by different water masses that are supposed to originate both from the north and from the south. The authors present a large amount of data that makes it sometimes very hard for the reader to follow the arguments because many data are only shown in the supplements but are discussed in the main text in length and are sometimes very important for the interpretation of the data. Here, it would be good to have some more information in the main figures (maybe one or two additional figures. Overall, I think that this is a nice contribution that is worth to be published in Climate of the Past. However, there are quite a few points that should be clarified by the authors to increase readability of the text and the data

interpretation. Furthermore, available datasets from the sites studied should be taken into account by the authors to support their statements and interpretation. Therefore, I recommend moderate to major revision.

My main points are (in the order they appear in the manuscript):

1) Abstract: why is the abbreviation for the Cretaceous WIS written with a K instead of a C?

2) The first paragraph and especially the first sentence of the introduction needs much more references to be included. Same is true for lines 20-30 on page 2.

3) line 11 on page 3: make clear that the benthic zone is only a small part of OAE 2.

4) line 11 on page 4: where is the connection between figures 1 and 2 and the text?

5) chapter 3.2.2 needs a reference to figure 8

6) chapter 3.3: at least present the most important features of the palyno-dataset that are used in the following discussion so that the reader has not to go back to the supplements every time.

7) line 25 on page 9: eigen scores are not shown in figures 7 and 11.

8) lines 29-31 on page 10: would delete this statement from the text.

9) lines 14-15 on page 11: since there is no increase in pollen and spores, there is no support for an increased hydrological cycle

10) line 19 on same page: what is the indication for climate cooling? Only the PCE is cooler, under background values that are much warmer than before or after OAE2!

11) Chapter 4.2.2: why should the tethyan water mass be suboxic-anoxic? This is inferred by the authors at the beginning of this chapter and then used in the following interpretation but it is never shown convincingly to the reader that this is the case. What is the independent evidence for this? Same holds true for the boreal water mass. Are

there any other indications other than own data and interpretations? If yes, present them in detail. So far, the main problem with this chapter is that there is no prove that the suggested water masses existed and are characterized by the suggested data in the way they are presented here.

12) line 18 on page 12: why is this indicative for an WIS source? Explain and justify in detail. Same with the argument in lines 28-29 with the shift from agglutinated to calcareous forams. Why is this a water mass characteristic and not simply a matter of preservation or changing food availability? Technically I wonder how the foram data were produced. Are the based on the linings in the palynological samples as indicated by the figure headings? I am not aware of a single study that has shown this to work.

13) line 31 on same page: at this point in the succession, there are no benthic forams, so the statement above in lines 28-29 cannot be valid!

14) lines 32ff on page 13: there are benthic foraminiferal assemblage data available form these sites, how do they compare to the data produced by linings? This would be a good test to show if the presented foram data of this study are of any value.

15) lines 2-3 on page 14: I am not aware that there are any data e.g. Nd isotopes from these sites that support a boreal influence. Furthermore, the authors state that this water mass should only influence shallow water settings. However, a cold boreal water mass should be denser than wormer waters near the tropics and therefore influencing bottom waters and not surface waters as suggested here.

16) line 35-36 on page 14: nowadays, nobody thinks anymore that the Cretaceous had a equitable climate!

17) lines 23-24 on page 15: but the red dots are all over the place in figure 15 and quite a few from Demerara Rise even above 3. What is the r2 for these data? Further in this chapter, Mo/TOC is used to say something about a silled basin situation at this site. Why not simply cite the papers that show that there was no sill during that time

(e.g. seismic evidence)?

18) last paragraph page 15: wouldn't be the absolute amount of refractory terrestrial organic matter (RTOM) an even more important factor than the T:M ratio alone? The ration could be high even when there is less RTOM and therefore a lower influence on Mo! Since this is not quantified, this is a weak justification and discussion.

19) first paragraph on page 16: How do these factors deplete Mo? Please explain the details.

20) lines 8-9 on same page: but isn't that what you are proposing above?

21) lines 21-23: check benthic assemblage data for these sites if available and see if there are benthic forams occurring in these intervals. If yes, these are oxygenation events, if not, it was anoxic. This would be an independent proof of the statements made here.

22) point 1 in the conclusions: are these water masses be interpreted to be surface and bottom-water masses at once? This has to be clarified in the discussion.

23) Figure 2: in part "a" it is hard to figure out the core locations.

24) Figure 3: what are the horizontal red lines?

---

## Author Response (AR1)

This file contains authors responses to review comments by **Anonymous Referee #1** and **Anonymous Referee #2** (responses R1.. in green); and marked-up manuscript

**Anonymous Referee #1**

The study by Eldrett and colleagues tries to understand and constrain the nature and timing of water-mass evolution in the southern gateway to the Cretaceous Western Interior Seaway (KWIS) and define the associated paleoenvironmental and paleoclimatic processes. This article presents detailed palynological and geochemical analyses from Cenomanian-Conacian interval from southwest Texas,USA. They also analysed materials from the central KWIS, to the south in the Tropical Atlantic and Southern Ocean in order to do correlations and reconstruct N-S water mass circulation. This work proposes another model of ocean circulation in the KWIS to explain the presence of normal oxygenation conditions during CIE (OAE-2) in this zone. It shows, inter alia, a link between sea level variations and N-S oceanic circulation reorganization. Finally, authorsdiscuss and try to explain the global trace metal drawdown during OAE-2.

This is an interesting article with many new data. It provides opportunities to discuss the impact of N-S water mass exchanges on regional paleoenvironmental changes during a period with a predominant latitudinal water mass circulation. However, many ortions/sections should be clarified or developed (- discussion about trace-metal significance (detrital versus authigenic origin); - discussion about sequential stratigraphy; - discussion about tethyan water mass in the western part of the Central Atlantic ocean: : :) and some paragraphs must be organised differently in order to help the reader (result and interpretation presentation with the same stratigraphic subdivision that is chronostratigraphic subdivision).

1. âˇA ´c p.4 ligne 36 : change "paleonenvironmental" by paleoenvironmental

R1. Thankyou for pointing this error, it is now corrected

2. âˇA ´c p.5 ligne 25 : "previously published organic and inorganic geochemical analyses" put bibliographic references

R2. We have inserted the appropriate bibliographic references as suggested.

3. âˇA ´c 2. Material and Method; In the supplementary data there is a lack of discussion on the authigenic origin of the trace metals used. To validate their types of source, the major and trace element abundances have to cross-correlated with Al abundances (indicators of detrital influx). See for example Lebedel et al., 2013 (Pal, Pal., Pal.)

R3. We had not included a discussion on the authigenic origin of the trace metals as we felt this had been addressed in previous publications.  We refer the reviewer to Eldrett et al. 2015b, Earth Planetary Science Letters 423, 98-113, in particular Figure 5 and discussion in section 3.3. The relevant trace metals and associated litho-facies are cross-correlated with Al and discussed in that paper. We have inserted a sentence into the supplementary information referring the readers to Eldrett et al. 2015b and have provided all data in the supplementary datafile where the data can be cross-plotted and interrogated.

4. You can also distinguish between redox proxies, Mo, V, U (Calvert and Pedersen, 1993) and palaeoproductivity proxies, Zn, Ni (Hatch and Leventhal, 1992).

R4. We do not believe that Zn and Ni can be distinguished from redox control and used solely as palaeo-productivity proxies. See also reply R11.

5. Finally, there is no Measurement accuracy.

R5. Measurement accuracy table added to the supplemental.

6. ă˘A ´c 3.1 Organic carbon-isotope Stratigraphy; In this section I think it is important to confront geochemical correlations with biostratigraphic correlations. This section seems to suggest that the identification of the Middle Cenomanian event and the Cenomanian-Turonian CIE is not based on age, but only on the magnitude of _13 Corg positive excursion.

R6. Clarification statement added. Carbon isotope stratigraphy is not just based on magnitude of the positive excursions, but also constrained by biostratigraphy (nannofossils, foraminifera, palynology), geochronology and astrochronologic calibration. These details were presented in Eldrett et al. 2015a and a clarification statement referring the readers to that publication for detailed discussion is inserted.

7. ă˘A ´c 3.2 Geochemistry p.6; It is necessary to re-organize this section. Use chronostratigraphic subdivision instead of lithostratigraphic subdivision. It lacks a description of the diverse sedimentary series that is the description of the main facies because the paleoenvironmental perturbations are recorded in the litho and biofacies. Are there cherts in these series?

R7. Re-organizing this section according to chronostratigraphy would in the authors opinion create unnecessary complexity and confusion. The results and majority of the text is presented in lithostratigraphic units and in depth as a presentation of results per core/outcrop section. Most of the geochemical and environmental signals are reflected in the deposition of lithology, not age. Therefore, presenting in chronostratigraphy, particularly the PCA results would be entirely confusing as the main lithological trends would be obscured. The extra interpretative step of regional trends in the discussion of water-masses is presented in chronostratigraphic age as an attempt at synthesis of the regionally important trends. To help clarify, age assigments are provided next to the associated depth in core (in parenthesis) so that the reader can orientate themselves in age as well as depth; also being consistent with the figures. We feel the main lithologic units are adequately presented in this contribution with focus on larger scale trends. Detailed environmental trends related to lithofacies (limestone-marlstone and bentonites) is presented by Eldrett et al. 2015b (Earth Planetary Science Letters 423, 98-113).bed-scale lithofacies descriptions are beyond the scope and is presented by Minisini et al. (Sedimentology, in review).

8. ă˘A ´c 3.3 Palynology p.7; In order to help the reader, it's better to present the meaning of the different parameters studied and presented in figures 4-10 (for example T/M ratio, Dinocyst P/G ratio, Shannon-Wiener diversity and Simpson Hunter Diversity). This information should notonly appear in the supplementary data.

R8. The meaning of the parameters are now presented in the methods (page 5)

9. ă˘A ´c 4.1 ´n Principal Component Analyses Â˙z (p. 8), Use chronostratigraphic subdivision instead of lithostratigraphic subdivision (same subdivision as geochemistry section).

R9. The authors see this would add complexity and confusion obscuring the main environmental signals. These are results and not presented in an interpreted chronostratigraphbic context. We have amended the text from 'stratigraphically' to 'lithostratigraphically'; ln. 25, and inserted, "The groups are discussed in a chronostratigraphic context in Section 4.2."

10. ∘A´c p.8 line 19 "Gallium (Ga)-Al2O3 [kaolinite]" I don't understand the direct link between Al2O3 and Kaolinite. Illite also contains Al2O3

R10. Gallium is generally associated with Al2O3 associated with detrital clays and agree that the direct link with kaolinite is difficult to establish.

11. ∘A´c p. 8 line 27 Âˈn redox sensitive trace metal concentrations Âˈz. Zn and Ni are also paleoproductivity proxies

R11. Although we agree with the reviewer in part, the main paleo-productivity elemental proxies are Barium and Phosphorus which have only one oxidation state. Zn, Cu, Ni are redox sensitive elements that are thought to be delivered to the sediment-water interface through the sinking of organic matter and could be used as proxies of organic matter flux. However, solubility and oxidation state of these elements are still a function of redox state and thus not solely an organic matter flux proxy. The delivery mechanisms and the calibration to surface water productivity are debated, so due to these uncertainties we have not split the element preferences from the primary category of redox sensitive. We retain the use of redox sensitive as it encompasses the main elements discussed in the paper (i.e. Mo, V, U).

12. ∘A´c p.8 line 35 change "paleonenvironmental" by paleoenvironmental

R12. We do not see this typo in the text.

13. ∘A´c p. 9 lines 19- 23 "In addition, it is interesting to note that although phytoclasts plot negatively along eigen axis 2 and may represent a reduced masking effect of AOM during oxygenated conditions (Tyson, 1995); they also plot positively along Eigen score 1 (noncarbonate/ volcaniclastic trend) alongside freshwater algae and Areoligeracean dinocysts suggestive of a more nearshore environment (Brinkhuis and Zachariasse 1988; Harker et al. 1990; Li and Habib 1996)". This last interpretation must be confirmed by the calculation of a correlation coefficient which, in my opinion, will not show a correlation. Here, you do only a suggestion but not a real interpretation because theposition of the phytoclasts is not at all correlated to the axis 1

R13. We agree with the reviewer. The suggestion was that freshwater algae and areoligeracean dinocysts are suggestive of a more nearshore environment by Brinkhuis and Zachariasse, 1988, Harker et al. 1990 and Li and Habib, 1996 rather than the relationship solely with axis 1. The statement has been amended to clarify the suggestion. It remains a suggestion as we state and believe other factors that we state (such as the masking effect and potential recycling) would complicate the relationship along axis 1 and reduce any correlation co-efficient.

14. ∘A´c p. 9 line 25 change "Figures 4-11" by Figures 4-6, 8-10

R14. Done, thankyou for the observation.

15. ∘A´c p.9 "Paleoenvironment Interpretation" Use the same chronostratigraphic subdivision as geochemistry section (3.2); If the sequence stratigraphy interpretation was not published before it's important to explain it.

R15. Chronostratigraphic age assignment for each of the main lithostratigraphic units presented and discussed in the text is provided in parentheses and we believe this as the optimal structure of the text. The sequence stratigraphic interpretation has not been previously published and is now included in the supplemental information.

16. ∘A´c p.11, line 16-19 "The sporomorph assemblages during OAE-2 mainly record a relative increase in gymnosperms, in particular during the PCE interval, and thus any increase in T:M ratio may reflect transition from mega-thermal to meso-thermal vegetation (perhaps also reflecting increased pollen production by wind dispersed gymnosperms) in response to

climate cooling episode rather than increased hydrologic cycle". Ti/Al can be used as a proxy for eolian versus fluvial input. What does this proxy show?

R16. The Ti/Al ratio shows an increase during the Plenus Marl event corresponding with recorded increase in gymnosperms. However, during this interval there is also an increase in mafic trace metal abundances, REE and Eu anomaly perhaps reflecting an igneous source, such as the High Arctic Large Igneous Province (see Eldrett et al. 2014; and supplemental figures). The increase in Ti/Al may reflect the emplacement and weathering of a LIP as well as alteration of biotite from the ubiquitous felsic volcanic ash beds rather than solely eolian versus fluvial inputs. We have included a sentence to discuss these uncertainties in Ti/Al as a proxy in the text and the relationship to increased gymnosperms and the PCE.

17. ☐A´c P.11 "Regional water-mass evolution"; "In this study we infer three main water-mass properties: i) a restricted suboxic-anoxic marine water-mass characterized by low diversity dinocyst assemblages interpreted to represent a tethyan source; ii) an unrestricted/open marine oxygenated water-mass characterized by high diversity dinocyst assemblages interpreted to represent a boreal source and iii) a partially restricted dysoxic water-mass interpreted to represent a more local central KWIS source."
It's too direct, you have to explain!! Why do you talk about a tethyan source and not an Atlantic-tethyan source? You specify ´n a restricted suboxic-anoxic marine watermass of tethyan source" before and after CIE. However Tethyan marine waters are generally well-oxygenated during these time intervals which are not the case of the Atlantic marine waters (see ref Monteiro et al., 2012 Paleoceanography for example: : :)". It would be better to write water mass of Atlantic-Tethyan source. Section organization: use the same chronostratigraphic subdivision than previously.

R17. This sentence is direct, but we feel appropriate. The reviewer is correct in that the definition of tethyan source was loosely applied here as part of the eastern Tethys was oxygenated during this time. As suggested we have amended the text and specifically refer to an equatorial-Atlantic tethyan source

18. ☐A´c p. 12 lines 8-12 "Lower Cenomanian sediments from the Equatorial Atlantic (ODP Site 1260) are interpreted as being deposited in a stratified suboxic-anoxic marine environment as indicated by laminated organic rich mudrock deposition and positive PCA-2 scores. This interpretation is consistent with a southern tethyan water-mass and a circulation controlled nutrient trap fuelling surface water productivity and anoxic depositional environment (Jiménez Berrocoso et al. 2010; Trabucho-Alexandre, 2010)". I don't understand why it's consistent with a southern tethyan water-mass. According Trabucho-Alexandre et al., 2010, this zone is the seat of upwelling of deep waters coming from the Pacific, no southern tethyan water-mass is mentioned by these authors.

R18. We agree with the reviewer. Our findings are consistent with the depositional conditions presented by Trabucho-Alexandre, 2010 and Jiménez Berrocoso et al. 2010 and not necessarily linked with a southern tethyan water mass. We have amended the text accordingly.

19. ☐A´c p.12 line 17 : "with mixed dinocyst assemblages". List of the genera

R19. We have amended the text to read: " In the Central KWIS (Portland-1 core), the early Cenomanian is characterized by relatively dysoxic depositional conditions, with frequent to common prasinophyte phycomata and mixed dinocyst assemblages, including taxa more typical of the northern and central KWIS such as *Senoniasphaera microreticulata* and *Palaeoperidinium cretaceum*, and the first consistent but mainly rare occurrence of *Bosedinia* cf. sp 1 & 3 which is common in coeval deposits at Demerara Rise; along with low diversity agglutinated benthic foraminfera (**Figure 8**) and the occasional occurrence of rare tethyan calcareous planktonic taxa, together suggestive of a mainly Western Interior Seaway source (Eicher and Diner, 1985)."

20. ăˇA ´c p.12, line 28 "In the Portland-1 core there is a clear shift from agglutinated to calcareous benthic foraminifera near the top of the MCE interval suggestive of a tethyan influence" why? Is there no agglutinated and calcareous benthic foraminifer in shallow water environment in the Atlantic and Pacific oceans?

R20. We agree with the comment of the reviewer and have clarified this sentence. The transition from agglutinated to calcareous foraminifera had been interpreted as reflecting the incursion of carbonate rich tethyan water into the KWIS, which in part is supported by occurrence planktonic foraminifera, ammonites, nannofossils of tethyan influence (see previous sentence). Detailed discussion of the foraminiferal assemblages we feel is beyond the scope of this contribution and have also inserted "for detailed discussion of tethyan-boreal foraminiferal distribution within Colorado, see Eicher and Diner, 1989)"

21. ăˇA ´c p.13, lines 35-39. "Furthermore, at ODP Site 1261, the recorded shift towards a more diverse and open-marine dinocyst assemblage is also associated with an increase in the abundance of organic foraminiferal test linings; re-population by calcareous benthic foraminifera (Friedrich et al., 2011), which combined with a reduction in redox sensitive trace metals is indicative of an improvement in environmental conditions and a reduction on the oxygen minimum zone". Warning ! There are few samples analysed; 4 samples in 10 m, it's little. This interpretation does not seem really justified because the organic matter concentration is very high.

R21. Indeed there are only four samples, however trace metal data and palynological data (including foraminiferal test linings) are from the same samples which show correspondence. In addition, much higher resolution is available from Friedrich et al. (2011) who recorded re-population of benthic foraminifera. Therefore, we believe it reasonable to suggest improved environmental conditions. However, the wording may be too strong given some of the uncertainty and potential for high frequency variations not captured in the sampling resolution. We have therefore amended the text from "is indicative" to "may indicate" and also inserted "However it should be noted that organic matter concentration remains high".

22. ăˇA ´c p.15 paragraph ´n Global trace metal draw-down during OAE-2" line 3, "During the Cenomanian, sediments that have been influenced by tethyan waters"

R22. Inserted "equatorial Atlantic"

**Anonymous Referee #2**

The study by Eldrett et al is a very nice contribution about the mid Cretaceous Western interior seaway and how WIS sediments are influenced by different water masses that are supposed to originate both from the north and from the south. The authors present a large amount of data that makes it sometimes very hard for the reader to follow the arguments because many data are only shown in the supplements but are discussed in the main text in length and are sometimes very important for the interpretation of the data. Here, it would be good to have some more information in the main figures (maybe one or two additional figures. Overall, I think that this is a nice contribution that is worth to be published in Climate of the Past. However, there are quite a few points

that should be clarified by the authors to increase readability of the text and the data interpretation. Furthermore, available datasets from the sites studied should be taken into account by the authors to support their statements and interpretation. Therefore, I recommend moderate to major revision. My main points are (in the order they appear in the manuscript):

1) Abstract: why is the abbreviation for the Cretaceous WIS written with a K instead of a C?

R1. This nomenclature for abbreviation follows conventional usuage and previous publications on the topic from the 1990's onwards; either shortening Cretaceous Western Interior Seaway to KWIS or Cretaceous Western Interior Basin to KWIB. As for the K-T boundary event the K is for Cretaceous and T for Tertiary; at the level of system/period, C is usually reserved for the Carboniferous. Within the Cretaceous, at the level of stage, C is used in this paper to abbreviate Cenomanian; if we abbreviated Cretaceous to C it may be confused with the C-T (Cenomanian-Turonian) boundary. The early workers of the Western Interior Seaway did not abbreviate; however to be more concise with fewer words we abbreviate so consistent with other publications.

2) The first paragraph and especially the first sentence of the introduction needs much more references to be included. Same is true for lines 20-30 on page 2.

R2. Our intention for the first sentence was to set the scene; with the detailed references relating to Large Igneous, sea level etc… being stated in lines 20-26. We agree with the reviewer that currently the first sentence reads light on the references, but our concern if inserted is that they will be duplicated in the following sentence. Lines2-30, page 2. We believe the key literature is referenced here; but will review available literature to assess whether additional references are warranted.

3) line 11 on page 3: make clear that the benthic zone is only a small part of OAE 2.

R3. We agree with the reviewer that the benthic zone as defined by Keller and Pardo (2004) is only a small part of OAE-2; however we also note that the original benthonic zone definition of Eicher and Worstell (1970) is much broader and in northerly KWIS localities spans the entire OAE-2 and post-OAE-2 interval. We have amended the text as follows:

"..tethyan water during OAE-2 was interpreted to have cumulated in the abrupt oxygenation of the seafloor as recorded by development and persistent abundance of benthic fauna (i.e. Elderbak and Leckie, 2016). However, this interval of benthic faunal abundance in the KWIS was originally termed the benthonic zone by Eicher and Worstell, (1970) who demonstrated that the benthonic foraminifera zone was best expressed in northerly sections where it spanned the entire Cenomanian-Turonian Bridge Creek Limestone, and is less developed in the central KWIS sections where it is stratigraphically restricted to the uppermost Cenomanian (i.e. beds 68-78 at Rock Canyon, Pueblo, Colorado; Eicher and Worstell, 1970), and where it spans only part of the OAE-2 interval and subsequently termed the Benthic Zone by Keller and Pardo (2004).

4) line 11 on page 4: where is the connection between figures 1 and 2 and the text?

R4. We agree with the reviewers comment. We have deleted "(Figures 1-2)" from the text and inserted reference to the figures more appropriately at the beginning of this section and in the methods. E.g. "The Eagle Ford Gr. was deposited during the Cenomanian to Turonian across the broad Comanche Platform in the southern intersection of the KWIS and northern Gulf of Mexico (Figure 1), "

5) chapter 3.2.2 needs a reference to figure 8

R5. We have inserted reference to Figure 8 as suggested.

6) chapter 3.3: at least present the most important features of the palyno-dataset that are used in the following discussion so that the reader has not to go back to the supplements every time.

R6. We utilized the supplementary information in order to reduce the size of the manuscript. We would like to refer to the editor whether the palynological results section should be re-instated to the main text.

7) line 25 on page 9: eigen scores are not shown in figures 7 and 11.

R7. This was also identified by reviewer #1; we have corrected.

8) lines 29-31 on page 10: would delete this statement from the text.

R8. We have deleted this comment

9) lines 14-15 on page 11: since there is no increase in pollen and spores, there is no support for an increased hydrological cycle

R9. In the sections from Texas and Demerara Rise there is not an increase in absolute abundance of pollen spores. However; in the Portland-1 core there is a slight increase (2,000- 5,000 c.p.g) so we disagree with the reviewers observation and a discussion is warranted. Regardless of absolute abundance, our data show a pollen assemblage shift to gymnosperm dominance; something that is highly relevant when comparing with other palynological records discussing increased hydrological cycle (e.g. Van Helmond et al. 2014) and subsequent citations.

10) line 19 on same page: what is the indication for climate cooling? Only the PCE is cooler, under background values that are much warmer than before or after OAE2!

R10. We agree with the reviewer that the primary evidence for climate cooling is linked with the PCE interval that elsewhere is associated with a sea surface temperature cooling (e.g. Forster et al. 2007; Van Helmond et al. 2014, 2016); climate cooling and drop in $_pCO^2$ (See Jarvis et al. 2011) as well as influx of boreal fauna. The increase in gymnosperms may therefore reflect expansion of conifer forests during this "cool snap". Additional evidence for climate cooling for the Turonian (i.e. what were background levels in the Turonian) are not well constrained. The persistence of boreal fauna in the sections presented here may indicate the presence of additional cooling episodes throughout the Turonian. Further work is required on this topic.

11) Chapter 4.2.2: why should the tethyan water mass be suboxic-anoxic? This is inferred by the authors at the beginning of this chapter and then used in the following interpretation but it is never shown convincingly to the reader that this is the case. What is the independent evidence for this? Same holds true for the boreal water mass. Are there any other indications other than own data and interpretations? If yes, present them in detail. So far, the main problem with this chapter is that there is no prove that the suggested water masses existed and are characterized by the suggested data in the way they are presented here.

R11. This point was also raised by reviewer #1. In our initial response to reviewer #1 were thought the introduction section was adequate/to the point. However, as both reviewers have raised this issue we have expanded the introduction and detail the independent evidence

12) line 18 on page 12: why is this indicative for an WIS source? Explain and justify in detail. Same with the argument in lines 28-29 with the shift from agglutinated to calcareous forams. Why is this a water mass characteristic and not simply a matter of preservation or changing food availability? Technically I wonder how the foram data were produced. Are the based on the linings in the palynological samples as indicated by the figure headings? I am not aware of a single study that has shown this to work.

R12. This point was also raised by reviewer #1. (comment 20 and response). As per previous response "We agree with the comment of the reviewer and have clarified this sentence. The transition from agglutinated to calcareous foraminifera had been interpreted as reflecting the

incursion of carbonate rich tethyan water into the KWIS, which in part is supported by occurrence planktonic foraminifera, ammonites, nannofossils of tethyan influence (see previous sentence). Detailed discussion of the foraminiferal assemblages we feel is beyond the scope of this contribution and have also inserted "for detailed discussion of tethyan-boreal foraminiferal distribution within Colorado, see Eicher and Diner, 1989)"

The benthic foram data was produced through two methods that we shall make clearer in the methods and figure captions; i) test linings from the palynological residues; ii) micropalaeontological analyses from sieved residues and thin sections. As previously mentioned, detailed inclusion of the micropalaeontological data we feel is beyond the scope of the paper and we primarily discuss the benthic abundances; inclusion would also lengthen the paper significantly as every sample analysed for palynology has an associated micropalaeontological dataset; as well as ~40 outcrops and we believe this requires a dedicated contribution. The occurrences of benthic foraminifera in the micropaleontological analyses and test linings in palynological assemblages are in general good agreement (as presented in Figure 4; quantified as c.p.g; and Figure 8 as relative abundance) and the method is demonstrated to work in this study. As below (comment 14) we shall also include published benthic foraminiferal data for sites 1260 and 1261.

13) line 31 on same page: at this point in the succession, there are no benthic forams, so the statement above in lines 28-29 cannot be valid!
R13. We disagree with the observation of the reviewer; there are foraminifera test linings (including for outside the initial 100 count specimens) in almost all samples throughout the Graneros to Hartland (with the exceptions of the 160.07m and 185.53m samples [column GQ in the Portland-1 datafile]). In addition, the statement above in lines 28-29 refers to the transition near the top of the MCE interval that is within the Graneros Shale and thus are also two completely different intervals of the core.

14) lines 32ff on page 13: there are benthic foraminiferal assemblage data available form these sites, how do they compare to the data produced by linings? This would be a good test to show if the presented foram data of this study are of any value.
R13. We agree with the reviewer and shall include benthic foraminifera abundances for Sites 1260 and 1261 in figures 9-10; and are in good agreement with the foram lining data.

15) lines 2-3 on page 14: I am not aware that there are any data e.g. Nd isotopes from these sites that support a boreal influence. Furthermore, the authors state that this water mass should only influence shallow water settings. However, a cold boreal watermass should be denser than wormer waters near the tropics and therefore influencing bottom waters and not surface waters as suggested here.
R15. As we state "The southern expression of this boreal influence is therefore limited in duration and extent". There is a Nd isotope excursion at this horizon and the nature of Nd signal is complex and requires additional localities from the KWIS., We state that our findings "indicating more complex interaction between water-masses and the oxygen minimum zone" and requires further work to resolve. It is an assumption by the reviewer that a boreal watermass would be denser as it would be colder; currently thermal gradients are not well constrained and salinity variations, particularly in the Western Interior Seaway are debated with suggestions of relatively freshwater (although we find no evidence of freshwater algae in our dataset). Our principal data are dinocysts that are mostly indicative of the photic zone so the watermasses are inferred to reflect surface water; however the vertical expression is apparent with sediment-water interface becoming oxygenated during OAE-2 associated with boreal taxa/watermass. Whether this is direct evidence for intermediate or deep water is unclear and we have included a statement in the text to raise this point – see also reviewer comment 22

16) line 35-36 on page 14: nowadays, nobody thinks anymore that the Cretaceous had a equitable climate!

R16. We agree and that is why it is stated .." than previously thought". In addition, this is an important point to make as not many studies have investigated this topic; the contribution of Gambacorta et al., (2016) is notable and our data is supportive of the proposition.

17) lines 23-24 on page 15: but the red dots are all over the place in figure 15 and quite a few from Demerara Rise even above 3. What is the r2 for these data? Further in this chapter, Mo/TOC is used to say something about a silled basin situation at this site. Why not simply cite the papers that show that there was no sill during that time (e.g. seismic evidence)?

R16. In unrestricted settings such as the Namibian Shelf and the OAE-2 interval presented here, there is a positive but generally weak co-variance between Mo and TOC and so would expect poor r2 values. In addition, preferential enrichment of Mo in euxinic conditions and in particular recycling in the particulate shuttle usually results in a TOC threshold for enhanced Mo enrichment; so the basic relationship is non-linear. We have clarified this in the text and inserted "weak positive co-variance between Mo and U..". Public domain evidence for absence of a sill across the US Gulf Coast is limited and along with the Demerara Rise ambiguous as reflect present-day geometries.

18) last paragraph page 15: wouldn't be the absolute amount of refractory terrestrial organic matter (RTOM) an even more important factor than the T:M ratio alone? The ration could be high even when there is less RTOM and therefore a lower influence on Mo! Since this is not quantified, this is a weak justification and discussion.

R18. We agree with the reviewers comment. The absolute amount of refractory terrestrial organic matter (RTOM) would be more significant than the T:M ratio alone. However; properly quantifying RTOM as far as we are aware is not currently possible. The major component of the organic matter is amorphous organic matter (AOM) and as stated in the text its origin is relatively poorly constrained; some part can be degraded terrestrial and/or marine in origin. Given that we cannot yet adequately identify and quantify the origin and composition of AOM and thus RTOM; the T:M ratio is the best approximation of terrestrially derived palynomorphs (and thus a component of RTOM). The text is slightly amended to include "refractory" and feel this uncertainty is now adequately captured, "We cannot determine the impact of this observation as the T:M ratio reflects a relatively small proportion of the total *refractory* organic matter, however further investigation is warranted into the variable origin of the more dominant and relatively unknown component, namely AOM; and whether Mo is preferentially incorporated within different organic matter components".

19) first paragraph on page 16: How do these factors deplete Mo? Please explain the details.

R19. The processes and controls have been included as suggested by the reviewer.

20) lines 8-9 on same page: but isn't that what you are proposing above?

R20. In part we agree, but the actual relationship between refractory organic matter and Mo is not documented by Dickson et al. (2016); as such this is the first integrated data of terrigeneous organic matter and Mo enrichments.

21) lines 21-23: check benthic assemblage data for these sites if available and see if there are benthic forams occurring in these intervals. If yes, these are oxygenation events, if not, it was anoxic. This would be an independent proof of the statements made here.

R21. Benthic foraminifera are present (Friedrich et al. 2006, 2011) and reflect re-population events associated with improved oxygenation. This data will be included in Figures 9-10 (see comment 14) and the following text has been inserted ". This interpretation is also supported by the occurrence of benthic foraminifera during the OAE-2 interval in both organic rich and organic lean sediments from

Demerara Rise (Friedrich et al. 2006, 2011); Texas (Lowery et al. 2014; Dodsworth, 2016; this study) and central KWIS (e.g. Eicher and Worstell, 1970; Keller and Pardo, 2004; Elderbeck and Leckie, 2016 and references therein).

22) point 1 in the conclusions: are these water masses be interpreted to be surface and bottom-water masses at once? This has to be clarified in the discussion.
R22. See comment R15. This has been clarified in the discussion.

23) Figure 2: in part "a" it is hard to figure out the core locations.
R23. Core locations shall be made bigger and unique symbology

24) Figure 3: what are the horizontal red lines?
R24. Horizontal red wavy lines are hiatal surfaces.  The captions shall be expanded to better explain the symbology in the figure

[revised manuscript text omitted]

**Commented [3]:**
Anonymous Referee #2 3
line 11 on page 3: make clear that the benthic zone is only a small part of OAE 2.

Clarified

Kump and Slingerland (1999), whereby a strong cyclonic gyre developed in the central KWIS drawing both tethyan waters northward along the eastern margin and boreal waters southward along the western margin of the seaway (see also discussions in Elderbak and Leckie, 2016).

5      In order to better understand and constrain the nature and timing of water-mass evolution in the southern gateway to the KWIS, and the associated paleoenvironmental and paleoclimatic processes, this contribution presents detailed palynological and geochemical analyses within a multidisciplinary framework from the Eagle Ford Group (Gr.) and bounding formations of the Buda Limestone and Austin Chalk from southwest Texas, USA. Furthermore, to place the southern gateway of the KWIS into a more supra-regional understanding we also

10    analysed correlative materials from the central KWIS (Portland-1 core, Colorado) and to the south in the Tropical Atlantic (ODP Leg 207, Demerara Rise) and Southern Ocean (ODP Leg 183, Kerguelen Plateau).

**1.1. Geological Setting**

The Eagle Ford Gr. was deposited during the Cenomanian to Turonian across the broad Comanche Platform in

15    the southern intersection of the KWIS and northern Gulf of Mexico (Figure 1), which represented part of the >3,000 km long foreland basin that formed behind the greater Cordilleran retro-arc fold-and-thrust belt during Late Mesozoic through Eocene times along the inboard side of the Cordilleran magmatic arc and accreted allochthonous terranes of North America (e.g., Burchfiel et al. 1992; Dickinson, 2004). The regional tectonic setting was influenced during the Cretaceous by the subduction of the conjugate oceanic plateau to the Shatsky

20    Rise (Liu et al., 2010) dynamic topography from the subducting Farallon Plate (Liu, 2014) and thermally subsiding Gulf of Mexico margin, resulting in the development of broad ramp-shelves including the Comanche Platform with reactivated basement structures defining intra-shelf basins, such as the Maverick Basin and structural highs such as the Terrell Arch. The Cenomanian-Turonian Trans-Pecos region (Figure 2) was deposited in a distal sediment starved setting >500 km from the nearest shoreline during a locally quiescent

25    tectonic period resulting in stable platform conditions and gradual subsidence ideal for the preservation of mudstones, limestones and bentonites of the Eagle Ford Gr., the underlying Buda Limestone and overlying Austin Chalk. (Figures 1-2).

> **Commented [4]:**
> Anonymous Referee #2 4
>
> line 11 on page 4: where is the connection between figures 1 and 2 and the text?
> R4. We agree with the reviewers comment. We have deleted "(Figures 1-2)" from the text and inserted reference to the figures more appropriately at the beginning of this section and in the methods.

[revised manuscript text omitted]

**Commented [6]:**
Anonymous Referee #1: 2
"previously published organic and inorganic geochemical analyses" put bibliographic references

DONE

**Commented [7]:**
Anonymous Referee #1: 8
Palynology p.7; In order to help the reader, it's better to present the meaning of the different parameters studied and presented in figures 4-10 (for example T/M ratio, Dinocyst P/G ratio, Shannon-Wiener diversity and Simpson Hunter Diversity). This information should notonly appear in the supplementary data.

DONE

and three negative- carbon isotope excursions (CIE) of varying magnitudes were recognized.  These excursions can be correlated with the English Chalk reference section of Jarvis et al. (2006; see Eldrett et al. 2015a for detailed discussion of biostratigraphic calibration of CIE's in the KWIS) and compared with the previously published data for ODP Leg 207 and KWIS as presented in **Figure 3**.  Two of these CIE's have specific relevance for this contribution: i) the ~2‰ positive $\delta^{13}C_{org}$ excursion in Iona-1 (143.73m-139.27m) and Innes (76.88m-74.63m) that corresponds with the Middle Cenomanian Event (MCE) and ii) the Cenomanian-Turonian CIE that is clearly expressed with a positive CIE of up to 4‰ occurring in Iona-1 (112.45 05.96 m-92.73m), Innes-1 (55.74 m- 42.51 m) and well 'X' (1639.9 m – top not recorded as above the cored interval). It should be noted that the definition of the base of the Cenomanian-Turonian CIE has been re-interpreted to include the precursor events presented in Eldrett et al. (2014, 2015a); as such the base of the CIE at Iona-1 is moved from 105.96m to 112.45m and is assigned an age of 95.01±0.12 Ma based on the obliquity age model presented in Eldrett et al. (2015).

**Commented [8]:**
Anonymous Referee #1. 6:
Organic carbon-isotope Stratigraphy; In this section I think it is important to confront geochemical correlations with biostratigraphic correlations. This section seems to suggest that the identification of the Middle Cenomanian event and the Cenomanian-Turonian CIE is not based on age, but only on the magnitude of _13 Corg positive excursion.

Clarification added

**3.2. Geochemistry**

**3.2.1. Southern KWIS (Texas)**

[revised manuscript text omitted]

**Commented [11]:**
Anonymous Referee #1. 10

"Gallium (Ga)-Al2O3 [kaolinite]" I don't understand the direct link between Al2O3 and Kaolinite. Illite also contains Al2O3.

Gallium is generally associated with Al2O3 associated with detrital clays and agree that the direct link with kaolinite is difficult to establish.

**Commented [12]:**
Anonymous Referee #1. 11
redox sensitive trace metal concentrations Âˊz. Zn and Ni are also paleoproductivity proxies

Reply:
Although we agree with the reviewer in part, the main paleo-productivity elemental proxies are Barium and Phosphorus which have only one oxidation state. Zn, Cu, Ni are redox sensitive elements that are thought to be delivered to the sediment-water interface through the sinking of organic matter and could be used as proxies of organic matter flux. However, solubility and oxidation state of these elements are still a function of redox state and thus not solely an organic matter flux proxy. The delivery mechanism and the calibration to surface water productivity is debated, so due to these uncertainties we have not split the element preferences from the primary category of redox sensitive. We retain the use of redox sensitive as it encompasses the main elements discussed in the paper (i.e. Mo, V, U).

The high negative eigen scores along axis 2 correspond with proxies associated with open marine oligotrophic conditions such as gonyaulacoid dinocysts (G-cysts) including *Spiniferites* spp., and *Pterodinium* spp.; the latter shows affinity with modern-day *Impagadinium* spp. which is found in oceanic and open marine conditions (Wall et al. 1977; Zonneveld et al. 2013). High negative eigen scores are also associated with proxies indicative of

5 more oxygenated water conditions such as higher concentration of the redox sensitive trace metal Manganese Oxide (MnO) and foraminiferal test linings. The negative eigen scores are therefore interpreted as representing an open marine, oxygenated and oligotrophic depositional environment.   In addition, it is interesting to note that although phytoclasts plot negatively along eigen axis 2 and may represent a reduced masking effect of AOM during oxygenated conditions (Tyson, 1995); they also plot positively along Eigen score 1 (non-

10 carbonate/volcaniclastic trend) alongside freshwater algae and Areoligeracean dinocysts; the latter are more suggestive of a more nearshore environment (see Brinkhuis and Zachariasse 1988; Harker et al. 1990; Li and Habib 1996).  In addition, a component of this trend may also represent recycling and transportation of nearshore palynomorphs to relativelymore distal environments. The eigen scores are plotted against depth for each of the sites in (**Figures 4-11 6, 8-10**) to further support the paleoenvironmental interpretation as discussed below.

**4.2.    Paleoenvironmental Interpretation**

**4.2.1.    Southern KWIS**

The Buda Limestone (early Cenomanian; ca 98-97.5 Ma) comprises highly diverse and low abundance dinocyst assemblages (see also Cornell, 1997) indicative of open marine oligotrophic conditions. This interpretation is

20 supported by the i) PCA results reflecting the lack of enrichment in redox sensitive trace metals, low TOC and poor preservation of AOM; dominance of G-cysts and ii) sedimentological and micropaleontological evidence for abundant and diverse benthos indicative of a healthy carbonate factory. The Buda Limestone dinocyst assemblages are comparable to those reported from marine limestone facies of early Cenomanian age in western Europe, including England (e.g. Cookson and Hughes, 1964) and France (e.g. Foucher, 1980).

The overlying lower Eagle Ford Gr. interval (ca. 97.2-94.9 Ma) is generally characterized by a decline in dinocyst species diversity, with the palynological assemblages comprising high absolute abundance of prasinophyte phycomata with P-cysts being the major component of the dinocyst community indicative of eutrophic and stratified water column conditions (cf. Prauss, 2007; Sluijs et al. 2005). This interpretation is

30 supported by PCA results reflecting the enrichment and co-variance in redox sensitive trace metals and high TOC values combined with low, but sporadic occurrences of benthic foraminifera indicative of restricted and suboxic-anoxic depositional conditions (see also Eldrett et al. 2014). Furthermore, the presence of aryl isoprenoids in the lower Eagle Ford Gr. section from Iona-1 has been demonstrated as originating from *Chlorobi* (green sulphur bacteria) and thus evidence for at least temporary and/or partial photic zone euxinia at this time

35 (Sun et al. 2016).  Bed- scale variations are also identified in redox conditions recording greater water-mass ventilation and current activity during the deposition of limestone beds compared to deposition of marlstone beds due to combined obliquity and precessional forcing on solar insolation (Eldrett et al. 2015b). Comparable

**Commented [13]:**
Anonymous Referee #1. 13

"In addition, it is interesting to note that although phytoclasts plot negatively along eigen axis 2 and may represent a reduced masking effect of AOM during oxygenated conditions (Tyson, 1995); they also plot positively along Eigen score 1 (noncarbonate/ volcaniclastic trend) alongside freshwater algae and Areoligeracean dinocysts suggestive of a more nearshore environment (Brinkhuis and Zachariasse 1988; Harker et al. 1990; Li and Habib 1996)". This last interpretation must be confirmed by the calculation of a correlation coefficient which, in my opinion, will not show a correlation. Here, you do only a suggestion but not a real interpretation because the position of the phytoclasts is not at all correlated to the axis 1

Text amended so clarified and response detailed

**Commented [14]:**
Anonymous Referee #1. 14
Anonymous Referee #2. 7
c p. 9 line 25 change "Figures 4-11" by Figures 4-6, 8-10

Done

**Commented [15]:**
Anonymous Referee #1. 15
c p.9 "Paleoenvironment Interpretation" Use the same chronostratigraphic subdivision as geochemistry section (3.2); If the sequence stratigraphy interpretation was not published before it's important to explain it.
Reply:
Chronostratigraphic age assignment for each of the main lithostratigraphic units presented and discussed in the text is provided in parentheses and we believe this as the optimal structure of the text.  The sequence stratigraphic interpretation has not been previously published and is now included in the supplemental information.

[revised manuscript text omitted]

**Commented [17]:**
Anonymous Referee #2. 9

lines 14-15 on page 11: since there is no increase in pollen and spores, there is no support for an increased hydrological cycle
R9. In the sections from Texas and Demerara Rise there is not an increase in absolute abundance of pollen spores. However; in the Portland-1 core there is a slight increase (2,000- 5,000 c.p.g) so we disagree with the reviewers observation and a discussion is warranted. Regardless of absolute abundance, our data show a pollen assemblage shift to gymnosperm dominance; something that is highly relevant when comparing with other palynological records discussing increased hydrological cycle (e.g. Van Helmond et al. 2014) and subsequent citations.

**Commented [18]:**
line 19 on same page: what is the indication for climate cooling? Only the PCE is cooler, under background values that are much warmer than before or after OAE2!
References added; also added TEX86 to Figure 13

**Commented [19]:**
Anonymous Referee #1. 16
Ti/Al can be used as a proxy for eolian versus fluvial input. What does this proxy show?

The Ti/Al ratio shows an increase during the Plenus Marl event corresponding with recorded increase in gymnosperms. However, during this interval there is also an increase in maffic trace metals, REE and recorded Eu anomaly interpreted as reflecting a maffic source, possible High Arctic Large Igneous Province (see Eldrett et al. 2014; and supplemental figures). The increase in Ti/Al may also contain a component or reflect the emplacement and weathering of a LIP rather than solely eolian versus fluvial input. Due to the uncertainty we have excluded this possible proxy in this publication.

**4.2.2. Regional water-mass evolution**

In this study we infer three main water-masses properties: i) eEquatorial-Atlantic tethyan source; ii) a northern boreal source and iii) a more local central KWIS source. The eEquatorial-Atlantic sourced water is primarily based on the occurrence of dinocysts and other microfauna (i.e. calcareous nannofossils and foraminifera) that are geographically restricted to low latitudes and specifically Eequatorial Atlantic with inferred tethyan affinities (e.g. *Bosedinia* cf. sp 1 & 3; see discussion below). The surface waters of this watermass is characterized by low diversity dinocyst assemblages dominated by P-cysts and prasinophyte phycomata indicative of stratified and hydrographically restricted and eutrophic conditions (this study); whilst the correspondence with the presence of isorenieratene derivatives (i.e. Sinninghe Damsté and Köster, 1998; Kuypers et al. 2002; Kolonic et al. 2005; Van Bentum et al, 2009; Sun et al. 2016) demonstrate at least temporary and/or partial photic zone euxinia. The underlying bottom-water; or at least sediment-water interface was predominantly dysoxic-anoxic as evidenced by the enrichment in redox sensitive trace metals and relatively sparse benthic foraminifera (i.e. Hetzel et a. 2009; Eldrett et al., 2014). The northerly sourced watermass is constrained by the occurrence of dinocysts with a boreal affinity (see Eldrett et al. 2014; Van Helmond et al. 2014; 2016); combined with the increased planktonic diversity and dominance of G-cysts that are associated with more oligotrophic/hydrographically unrestricted conditions. The absence of prasinophyte algae point towards a mixed water-column (cf. Prauss, 2007); and the overall low values in redox sensitive elements and increased abundance and diversity of benthic fauna along with enhanced bioturbation indices (e.g. Meyers et al. 2007; Eldrett et al. 2014) indicate frequent ventilation of the bottom-waters and/or sediment-water interface. A more local and partially restricted dysoxic watermass sourced by the central KWIS is interpreted based on the occurrence of mixed moderate diversity dinocyst assemblages, including taxa more typical of the northern and central KWIS, with rare tethyan components combined with limited enrichment in redox sensitive elements and abundance of low diversity agglutinated benthic foraminfera assemblages (see discussion below). It should be noted that within these regional watermass regimes; higher frequency variations in redox state are documented; in part a response to obliquity and precession forcing on the latitudinal distribution of solar insolation (see Eldrett et al. 2015b). a restricted suboxic-anoxic marine water-mass characterized by low diversity dinocyst assemblages interpreted to represent a tethyan source; ii) an unrestricted/open marine oxygenated water-mass characterized by high diversity dinocyst assemblages interpreted to represent a boreal source and iii) a partially restricted dysoxic water-mass interpreted to represent a more local central KWIS source. Comparing the paleoenvironmental trends recorded in the Cenomanian – Coniacian sections from SW Texas to those recorded further north within the central KWIS (USGS Portland-1; Pueblo GSSP outcrop) and to the south in the equatorial North Atlantic (ODP sites 1260, 1261, Demerara Rise) allows the reconstruction of their water-mass evolution allows an attempt at reconstructing the evolution of these various water-masses (**Figure 13-14**). A discussion of dinocyst and pollen paleo-latitudinal provinicialism across these regions during Cenomanian and Turonian times is given in the supplemental information.
* * *
**Commented [20]:**
Anonymous Referee #1. 17
It's too direct, you have to explain!! Why do you talk about a tethyan source and not an Atlantic-tethyan source? You specify ´n a restricted suboxic-anoxic marine watermass of tethyan source" before and after CIE. However Tethyan marine waters are generally well-oxygenated during these time intervals which are not the case of the Atlantic marine waters (see ref Monteiro et al., 2012 Paleoceanography for example: : :)". It would be better to write water mass of Atlantic-Tethyan source. Section organization: use the same chronostratigraphic subdivision than previously.

As suggested we have amended the text and specifically refer to an equatorial-Atlantic tethyan source

Eldrett, James S GSNL-PTI/EG

Anonymous Referee #2. 11

11) Chapter 4.2.2: why should the tethyan water mass be suboxic-anoxic? This is inferred by the authors at the beginning of this chapter and then used in the following interpretation but it is never shown convincingly to the reader that this is the case. What is the independent evidence for this? Same holds true for the boreal water mass. Are there any other indications other than own data and interpretations? If yes, present them in detail. So far, the main problem with this chapter is that there is no prove that the suggested water masses existed and are characterized by the suggested data in the way they are presented here.

ammended

**Commented [21]:**
Anonymous Referee #2. 11

11) Chapter 4.2.2: why should the tethyan water mass be suboxic-anoxic? This is inferred by the authors at the beginning of this chapter and then used in the following interpretation but it is never shown convincingly to the reader that this is the case. What is the independent evidence for this? Same holds true for the boreal water mass. Are there any other indications other than own data and interpretations? If yes, present them in detail. So far, the main problem with this chapter is that there is no prove that the suggested water masses existed

**Commented [22]:**
Anonymous Referee #1. 17

[revised manuscript text omitted]

**Commented [23]:**
Anonymous Referee #1. 18
c p. 12 lines 8-12 "Lower Cenomanian sediments from the Equatorial Atlantic (ODP Site 1260) are interpreted as being deposited in a stratified suboxic-anoxic marine environment as indicated by laminated organic rich mudrock deposition and positive PCA-2 scores. This interpretation is consistent with a southern tethyan water-mass and a circulation controlled nutrient trap fuelling surface water productivity and anoxic depositional environment (Jiménez Berrocoso et al. 2010; Trabucho-Alexandre, 2010)". I don't understand why it's consistent with a southern tethyan water-mass. According Trabucho-Alexandre et al., 2010, this zone is the seat of upwelling of deep waters coming from the Pacific, no southern tethyan water-mass is mentioned by these authors.

We agree with the reviewer. Our findings are consistent with the depositional conditons presented by Trabucho-Alexandre, 2010 and Jiménez Berrocoso et al. 2010

**Commented [24]:**
Anonymous Referee #1. 19
c p.12 line 17 : "with mixed dinocyst assemblages". List of the genera

We have amended the text

Eldrett, James S GSNL-PTI/EG
why is this indicative for an WIS source? Explain and justify in detail.

done

**Commented [25]:**
Anonymous Referee #2. 13
line 31 on same page: at this point in the succession, there are no benthic forams, so the statement above in lines 28-29 cannot be valid!

We disagree with the observation of the reviewer; there are foraminifera test linings (including for outside the initial 100 count specimens) in almost all samples throughout the Graneros to Hartland (with the exceptions of the 160.07m and 185.53m samples [column GQ in the Portland-1 datafile]).

**Commented [26]:**
Anonymous Referee #1. 20
c p.12, line 28 "In the Portland-1 core there is a clear shift from agglutinated to calcareous benthic foraminifera near the top of the MCE interval suggestive of a tethyan influence" why? Is there no agglutinated and calcareous benthic foraminifer in shallow water environment in the Atlantic and Pacific oceans?

Reply: We agree with the comment of the reviewer and have clarified this sentence. The transition from agglutinated to calcareous foraminifera had been

[revised manuscript text omitted]
". Warning ! There are few samples analysed; 4 samples in 10 m, it's little. This interpretation does not seem really justified because the organic matter concentration is very high.

Reply:
Indeed there are only four samples, however trace metal data and palynological data (including foraminiferal test linings) are from the same samples which show correspondence. In addition, much higher resolution is available from Friedrich et al. (2011) who recorded re-population of benthic foraminifera. Therefore, we believe it reasonable to suggest improved environmental conditions. However, the wording may be too strong given some of the uncertainty and potential for high frequency variations not captured in the sampling resolution. We have therefore amended the text from "is indicative" to "may indicate" and also inserted "However it should be noted that organic matter concentration remains high".

Eldrett, James S GSNL-PTI/EG

Anonymous Referee #2. 14
lines 32ff on page 13: there are benthic foraminiferal assemblage data available
form these sites, how do they compare to the data produced by linings? This would be
a good test to show if the presented foram data of this study are of any value.
R13. We agree with the reviewer and shall include benthic foraminifera abundances for Sites 1260 and ...

**Commented [28]:**
Anonymous Referee #2. 15
lines 2-3 on page 14: I am not aware that there are any data e.g. Nd isotopes from
these sites that support a boreal influence. Furthermore, the authors state that this water mass should only influence shallow water settings. However, a cold boreal watermass should be denser than wormer waters near the tropics and therefore influencing bottom waters and not surface waters as suggested here

R15. As we state "The southern expression of this boreal influence is therefore limited in duration and extent". There is a Nd isotope excursion at this horizon and the nature of Nd signal is complex and requires additional localities from the KWIS., We state that our findings "indicating more complex interaction between water-masses and the oxygen minimum zone" and requires further work to resolve. It is an assumption by the reviewer that a boreal watermass would be denser as it would be colder; currently thermal gradients are ...

regional climatic and/or tectonic subsidence-induced incursions of boreal water that ventilated the seaway rather than solely by global eustatic drivers that are difficult to reconcile with a greenhouse world without significant polar ice. The development of coeval hiatal intervals are also recorded near the top and base of the OAE-2 CIE in the Bonarelli interval, Italy, where vigorous bottom currents were proposed to be induced by warm and dense

5 saline deep waters that originated on tropical shelves in the Tethys and/or proto-Atlantic Ocean (Gambacorta et al. 2016). Although our contribution proposes an alternative mechanism for invigorated circulation, it further supports the suggestion by Gambacorta et al. (2016) that enhanced current activity and associated oceanographic circulation was much more dynamic than previously thought during times of Greenhouse climates when conditions were thought to be more equitable.

**4.2.3. Global trace metal draw-down during OAE-2**

During the Cenomanian, sediments that have werebeen influenced by equatorial-Atlantic tethyan waters record redox sensitive trace metals that are enriched compared to average shale, in particular Molybdenum (Mo) which requires free $H_2S$ for authigenic sedimentary enrichment (-(Helz et al. 1996) and is interpreted as reflecting

15 anoxic to euxinic depositional conditions favouring organic matter preservation (see Algeo and Lyons, 2006). The relative enrichment of Mo compared to the other redox elements such as U and V in the pre-OAE-2 lower Eagle Ford Gr., suggests an active cycling of Mn–Fe particulates (especially Mn-oxyhydroxides) between the water column and sediment water interface characteristic of a "particulate shuttle" (Algeo and Tribovillard, 2009; Tribovillard et al. 2012; **Figure 15a**). The presence of aryl isoprenoids in this interval from the Iona-1

20 core also supports at least temporary and partial water column (photic zone) euxinia at this time (Sun et al. 2016).

During the onset of OAE-2 the recorded depletion of sedimentary redox sensitive trace metals has been proposed to reflect the draw-down of the global trace metal sea-water inventory due to sequestration in sediments under expanded anoxic/euxinic conditions (Hetzel et al. 2009; Dickson et al. 2016; Goldberg et al. 2016). These

25 interpretations are based primarily on the exceptionally low Mo/TOC gradients compared to thoseat documented in modern-day anoxic silled marine basins (Algeo and Lyons, 2006), whereby removal of aqueous Mo concentrations under anoxic/euxinic conditions to the sediment is in excess to resupply by inter-basinal transfer of water masses. This contribution also documents exceptionally low Mo/TOC gradients for sediments spanning the OAE-2 interval from Demerara Rise and SW Texas (**Figure 15b**; also see Hetzel et al. 2009; Eldrett et al.

30 2014 Fig. DR3), but also from those recovered from the upper Eagle Ford Gr., and from the Dakota Sandstone interval from the Portland-1 core (supplemental datafile).

However, there are three main challenges to the application of Mo/TOC gradients to infer global trace metal inventory drawdown during OAE-2. Firstly, during the OAE-2 interval in the KWIS, Demerara Rise and

35 Kerguelen Plateau, the weak positive co-variance between Mo and U is indicates d of deposition along the unrestricted open marine trend (**Figure 15a**; also Eldrett et al. 2014); an interpretation supported by the palynological assemblages presented here. Therefore, during the OAE-2 interval, these localities were not subject to significant hydrographic restriction (e.g., silled basins), being deposited instead along continental margins. Thus the application of modern-day Mo/TOC gradients to infer hydrographic restriction and trace metal

**Commented [29]:**
Anonymous Referee #2. 16

line 35-36 on page 14: nowadays, nobody thinks anymore that the Cretaceous had
a equitable climate!
R16. We agree and that is why it is stated .." than previously thought". In addition, this is an important point to make as not many studies have investigated this topic; the contribution of Gambacorta et al., (2016) is notable and our data is supportive of the proposition.

**Commented [30]:**
Anonymous Referee #2. 17

lines 23-24 on page 15: but the red dots are all over the place

We have clarified this in the text and inserted "weak positive co-variance between Mo and U..".

drawdown for OAE-2 may not be valid and requires additional study (see discussion in Algeo and Lyons, 2006).

Secondly, as Mo is mostly incorporated into organic matter, we propose that the various contributions of different types of organic matter would affect Mo/TOC gradients. Whereas sulfurized organic matter is known to enhance Mo uptake by the sediment (Tribovillard et al. 2004), the impact of variable contributions of labile versus refractory organic matter is not well constrained. The palynological analyses presented here demonstrate increased contribution of refractory terrigenous organic matter (>T:M ratio) in the samples with low Mo/TOC values from the KWIS and Demerara Rise (**Figure 15b**). We cannot determine the impact of this observation as the T:M ratio reflects a relatively small proportion of the total refractory organic matter, however further investigation is warranted into the variable origin of the more dominant and relatively unknown component, namely AOM; and whether Mo is preferentially incorporated within different organic matter components.

Thirdly; in all the sections presented here the depletion in redox sensitive trace metals during OAE-2 is either associated with oxygenated depositional conditions, or an improvement in redox state. Therefore, the recorded depletion in redox sensitive trace metals, in particular Mo may instead record a genuine environmental response to benthic oxygenation preventing authigenic sedimentary enrichments as under oxic conditions, Mo is present in seawater as the stable and largely unreactive molybdate oxyanion. These observations are interpreted to be related to the equatorial migration of boreal water-masses promoting water column de-stratification and partial ventilation/re-oxygenation of the surface -  deep waters.

These challenges were partly addressed by Dickson et al. (2016) who proposed that the recorded low Mo/TOC gradients from Site 1138, Kerguelen Plateau reflected the widespread and global nature of trace-metal depletion during OAE-2 by demonstrating that i) localized oxygenation was also not responsible due to Mo isotope compositions and ii) organic matter type was not a control due to the low abundance of terrigenous material. Palynological investigation of these sediments (this study) confirms the low abundance of terrigenous material at Site 1138.  In addition, the dinocyst assemblage recovered from this site is typical of a well-oxygenated boreal equivalent (austral) water-mass with high diversity and abundance of G-cysts, notably *Cyclonephelium compactum- membraniphorum* dinocyst plexus, and outer neritic to open marine taxa such as *Spiniferites ramosus* and *Pterodinium cingulatum*. These findings would suggest deposition under the influence of an oxygenated and open marine/hydrographically unrestricted austral (Southern Hemisphere) surface water-mass on the Kerguelen Plateau; consistent with the relatively low $\delta^{98/95}$Mo values and Mo-U co-variance trends reflecting oxic-suboxic depositional conditions; with at most sulphidic pore-waters (Dickson et al. 2016). Furthermore, conditions at the sediment-water interface were sufficient for the presence of a low diversity, high carbon-flux benthic foraminifera biofacies (Holbourn and Kuhnt, 2002) as also indicated by the rare occurrence of organic linings of benthic foraminifera throughout the OAE-2 interval at Site 1138 (this study). Therefore, in addition to trace metal drawdown as an explanation to reconcile low Mo/TOC gradients and hydrologically unrestricted regime.; it is possible that even in the organic-rich laminated sedimentary intervals (i.e. Sites 1138 and 1260) a slight increase in oxic conditions related to the influence of oxygenated high-latitude austral-boreal water-masses, may result in aqueous $H_2S$ concentrations at the sediment- water interface being below the

Commented [31]:
Anonymous Referee #2. 18
last paragraph page 15: wouldn't be the absolute amount of refractory terrestrial organic matter (RTOM) an even more important factor than the T:M ratio alone? The ration could be high even when there is less RTOM and therefore a lower influence on Mo! Since this is not quantified, this is a weak justification and discussion.

The text is slightly amended to include "refractory" and feel this uncertainty is now adequately captured

Commented [32]:
Anonymous Referee #2. 19

last paragraph page 15: wouldn't be the absolute amount of refractory terrestrial organic matter (RTOM) an even more important factor than the T:M ratio alone? The ration could be high even when there is less RTOM and therefore a lower influence on Mo! Since this is not quantified, this is a weak justification and discussion.

Text amended and processes added

Commented [33]:
Anonymous Referee #2. 20

lines 8-9 on same page: but isn't that what you are proposing above?
In part we agree, but the actual relationship between refractory organic matter and Mo is not documented by Dickson et al. (2016); as such this is the first integrated data of terrigeneous organic matter and Mo enrichments.

critical threshold for conversion of molybdate to thiomolybdate (Helz et al. 1996), resulting in the observed depletion in Mo sedimentary concentrations and low Mo/TOC gradients during OAE-2. This interpretation is also supported by the occurrence of benthic foraminifera within the OAE-2 interval in both organic rich and organic lean sedimentary sections from Demerara Rise (Friedrich et al. 2006, 2011); Kerguelen Plateau (Holbourn and Kuhnt, 2002); Texas (Lowery et al. 2014; Dodsworth, 2016; this study) and central KWIS (e.g. Eicher and Worstell, 1970; Keller and Pardo, 2004; Elderbeck and Leckie, 2016 and references therein).

**5. Conclusions**

This integrated palynological and geochemical study using a multidisciplinary approach has provided insights into depositional environments of the Cenomanian-Turonian Eagle Ford Gr. and bounding formations of the Buda Limestone and Austin Chalk from southwest Texas, USA. The study spans a physiographic transect across the Comanchean Shelf from the shallow setting of the San Marcos Arch (AU2); through the main shelf (Innes-1), the slope (Iona-1) to the intra-shelf basin (well X). Furthermore, comparison of the paleoenvironmental trends recorded in the Cenomanian – Coniacian section from SW Texas to those recorded further north within the central KWIS (USGS Portland-1) and to the south in the equatorial North Atlantic (ODP sites 1260, 1261, Demerara Rise) and Kerguelen Plateau (ODP Site 1138) allows the interpretation of the evolution of various water-masses along this north-south transect. The main findings are:

1. High latitude boreal and austral (Northern and Southern Hemisphere) and equatorial Atlantic tethyan water-masses can be distinguished based on their distinct palynological assemblages and geochemical signatures.

2. The northward flow of a suboxic-anoxic equatorial Atlantic tethyan water-mass into the Western Interior Seaway occurred during the early-middle Cenomanian; followed by a major re-organisation of the oceanographic regime during the latest Cenomanian- Turonian as a full connection of the Western Interior Seaway with a northerly- boreal water-mass was established during peak transgression. This oceanographic change promoted de-stratification of the water column and improved oxygenation throughout the KWIS and as far south as the Demerara Rise.

3. These long term trends in water-mass evolution are tentatively linked to third order eustatic transgression-regression cycles driven by regional Cordilleran tectonic and/or mantle plume-lithosphere dynamics associated with the emplacement of LIPs during this time as well as shorter term variations in climate (i.e. Plenus Cold Event).

4. Low Mo/TOC ratios in the equatorial North Atlantic in comparison to other oceanic basins during the onset of OAE-2 argue for partial restriction and draw-down of global trace metal sea-water inventories. However, this study demonstrates that the recorded decline in redox-sensitive trace metals during the onset of OAE-2 likely reflect either a genuine oxygenation event related to open water-mass exchange

**Commented [34]:**
Anonymous Referee #2. 21

lines 21-23: check benthic assemblage data for these sites if available and see if there are benthic forams occurring in these intervals. If yes, these are oxygenation events, if not, it was anoxic. This would be an independent proof of the statements made here.

Benthic foraminifera are present, text ammended

**Commented [35]:**
Anonymous Referee #2. 22
point 1 in the conclusions: are these water masses be interpreted to be surface and bottom-water masses at once? This has to be clarified in the discussion.
R22. See comment R15. This has been clarified in the discussion.

[revised manuscript text omitted]